# Dropout Methods for Bilevel Training Task

**Peiran Yu** *
Department of Computer Science
University of Maryland
College Park, MD 20740, USA
{pyu123}@umd.edu

**Junyi Li**
Department of Computer Science
University of Maryland
College Park, MD 20740, USA
{junyili.ai}@gmail.com

**Heng Huang** †
Department of Computer Science
University of Maryland
College Park, MD 20740, USA
{henghuanghh}@gmail.com

## Abstract

Bilevel optimization problems appear in many widely used machine learning tasks. Bilevel optimization models are sensitive to small changes, and bilevel training tasks typically involve limited datasets. Therefore, overfitting is a common challenge in bilevel training tasks. This paper considers the use of dropout to address this problem. We propose a bilevel optimization model that depends on the distribution of dropout masks. We investigate how the dropout rate affects the hypergradient of this model. We propose a dropout bilevel method to solve the dropout bilevel optimization model. Subsequently, we analyze the resulting dropout bilevel method from an optimization perspective. Analyzing the optimization properties of methods with dropout is essential because it provides convergence guarantees for methods using dropout. However, there has been limited investigation in this research direction. We provide the complexity of the resulting dropout bilevel method in terms of reaching an $\epsilon$ stationary point of the proposed stochastic bilevel model. Empirically, we demonstrate that overfitting occurs in data cleaning and meta-learning, and the method proposed in this work mitigates this issue.

## 1 Introduction

Bilevel optimization appear in many problems widely used in machine learning tasks such as data cleaning (Shaban et al., 2019), hyperparameter optimization, meta-learning (Franceschi et al., 2018) and reinforcement learning (Yang et al., 2019). The bilevel optimization problem involves two minimization problems that are stacked on top of each other, where the solution to one optimization problem depends on the solution of the other. Take data cleaning as an example. Suppose we have a corrupted training data set $\mathcal{D}_{tr}$ with $N_{tr}$ data points and a clean data set $\mathcal{D}_{val}$ with $N_{val}$ data points. Let $f(w;\xi)$ be a network parametrized by $w$. The aim of data cleaning is to train the model with corrupted data by solving

$$\min_{\lambda \in \mathbb{R}^{N_{tr}}} \frac{1}{|N_{val}|} \sum_{i=1}^{N_{val}} l(f(w(\lambda);\eta_i);\eta_i) \text{ s.t.} w(\lambda) \in \underset{w}{\text{Arg min}} \frac{1}{|N_{tr}|} \sum_{i=1}^{N_{tr}} \sigma(\lambda_i) l(f(w;\xi_i);\xi_i) \quad (1)$$

where $\eta_i \in \mathcal{D}_{val}$, $\xi_i \in \mathcal{D}_{tr}$, $l$ is the loss function and $\sigma(\lambda_i)$ is the weight of the loss w.r.t. the data $\xi_i$. If $\xi_i$ is corrupted, we hope $\sigma(\lambda_i)$ can neutralize the influence of $\xi_i$.

Another important machine learning task using bilevel optimization is the meta-learning (Franceschi et al., 2018; Lorraine et al., 2020b). Suppose we have $N$ different training tasks. The $j$th task

---

*Use footnote for providing further information about author (webpage, alternative address)—*not* for acknowledging funding agencies. Funding acknowledgements go at the end of the paper.
†

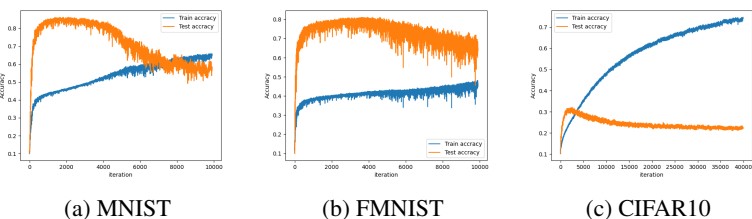

(a) MNIST          (b) FMNIST          (c) CIFAR10

Figure 1: Overfitting in data cleaning tasks

has a pair of training and validation data sets $\{(\mathcal{D}_{j,v}, \mathcal{D}_{j,tr})\}$. The $j$-th task is to train a network $f(w_j, \lambda; \xi)$, where $\xi$ is an input, $w_j$ denotes the parameters for task $j$ and $\lambda$ denotes the parameters shared for all $N$ tasks. The goal of meta-learning is to find good shared parameters $\lambda$. To this end, the following bilevel model is considered:

$$\min_{\lambda} \frac{1}{N} \sum_{j=1}^{N} l_j(w_j, \lambda; \mathcal{D}_{j,v}) \text{ s.t. } w(\lambda) \in \operatorname*{Arg\,min}_{w := (w_1, \dots, w_N)} \frac{1}{N} \sum_{j=1}^{N} l_j(w_j, \lambda; \mathcal{D}_{j,tr}), \qquad (2)$$

where $l_j(w_j, \lambda; \mathcal{D})$ is the loss of the network $f$ for the $j$th task on a data set $\mathcal{D}$.

As shown in (1) and (2), bilevel training tasks are more complex than single level training tasks. Thus, bilevel optimization models can be sensitive to changes in the input data and parameters. In addition, bilevel training tasks usually suffer from limited data sets. As evidenced by Figure 1, in the data cleaning problem, overfitting happens since the increase in the training accuracy leads to a decrease in the testing accuracy of the classifier. Regarding meta-learning, overfitting in the model learned through bilevel optimization can result in a diminished ability to learn new tasks. As shown in our experiments section, in classification problems, the training accuracy is high while the testing accuracy is not. Therefore, overfitting is a common challenge in bilevel training tasks and we need to addressing it.

In single level machine learning optimization problems such as empirical risk minimization, there are many ways to redeem the overfitting problem, including dropout, early stopping, data augmentation, etc. Among these methods, dropout is highly effective, simple to apply and computationally efficient, (Srivastava et al., 2014; Labach et al., 2019). Proposed by Srivastava et al. (2014), dropout randomly drops neurons of the neural network during training, leading to only a portion of parameters being updated in each epoch.Dropout has been successfully applied to fully connected networks (Ba & Frey, 2013; Srivastava et al., 2014), convolutional neural networks (CNNs) (Wu & Gu, 2015; Srivastava et al., 2014; Park & Kwak, 2017), and recurrent layers in recurrent neural networks (RNNs) (Pham et al., 2014; Zaremba et al., 2014). In this work, we investigate how dropout behaves when adopted to solving the bilevel optimization problem.

Since the dropout only randomly zero out part of the neurons in a network, we propose a bilevel optimization model that characterizes this randomness. Since current existing bilevel methods are not directly applicable to this new model, we use a representative bilevel method from Li et al. (2022) as an example to see how existing bilevel methods can be adapted to this new model. We then investigate theoretical convergence guarantees of the resulting dropout bilevel method. Existing analysis of the dropout method from an optimization perspective is very limited. As far as we search, only (Li et al., 2016; Mianjy & Arora, 2020; Senen-Cerda & Sanders, 2020; Senen-Cerda & Sanders, 2022) investigated this direction previously. However, these works only focused on single level optimization problems. For bilevel problems, there are no theoretical convergence guarantees for dropout methods. In this work, we fill in this gap and study the convergence properties for a dropout bilevel method. The challenges in analyzing the dropout bilevel method, which the single-level model considered in (Senen-Cerda & Sanders, 2020) does not face, include how to analyze the hypergradient. Analyzing the hypergradient is hard to analyze in two ways: 1. it is related to the solution of the lower level problem; 2. both upper and lower level objectives are composed with random dropout masks.

Our contributions are summarized as follows:

- We form a statistical bilevel optimization problem that includes the distribution of the dropout masks. To solve this problem, we propose a dropour variant of an existing method

from (Li et al., 2022) as an example of a bilevel method. We investigate the inductive relations between the variables that are used in the dropout bilevel method to approximate the hypergradients of the outer-level objective. These variables are affected by the dropout masks, and the variance term in the inductive relations is affected by the dropout rates.

- We show that the complexity of the dropout bilevel method depends on the dropout rate. Unlike the complexity of the bilevel methods for the bilevel model without dropout, the complexity of the dropout bilevel method for the dropout bilevel model has additional errors that are only brought by the dropout. The challenge in this analysis is how to handle the distributions of dropout masks that appear when estimating the error of the hypergradient of the upper-level objective.

- We apply the proposed method to data cleaning problems. In our experiments, we observe the overfitting problem. In addition, we observe that the proposed dropout bilevel method redeems the overfitting problem. We also observe that the dropout bilevel method converges. Furthermore, we observed that accuracy changes across iterations are more stable when using dropout.

## 1.1 RELATED WORK

Theoretical properties of dropout methods for single-level optimization problems have been investigated in (Baldi & Sadowski, 2013; Gal & Ghahramani, 2016; Zhai & Wang, 2018; Li et al., 2016). In particular, Baldi & Sadowski (2013) introduced a general formalism for studying dropout and use it to analyze the averaging and regularization properties of dropout. Gal & Ghahramani (2016) and (Gal & Ghahramani, 2016) formed dropout training as approximate Bayesian inference in deep Gaussian processes and studied the model uncertainty. (Zhai & Wang, 2018; Gao & Zhou, 2016; Wang et al., 2019) studied Rademacher complexity bound of the dropout method.

Li et al. (2016); Mianjy & Arora (2020); Senen-Cerda & Sanders (2020); Senen-Cerda & Sanders (2022) investigated the convergence of the dropout method for single level risk minimization tasks. Li et al. (2016) view dropout as a data-dependent regularizer added to the training loss. In contrast, our approach involves forming a stochastic minimization model that treats dropout masks as a random linear transformation within the training loss. Mianjy & Arora (2020) considered a dropout method for a 2-layers network. When the loss function is logistic loss (convex) and the activation function is Relu, they provided the convergence rate of testing error. In this work, we consider general multi-layer neural networks and investigate training error. In addition, they assume data distribution is separable by a margin in a particular Reproducing Kernel Hilbert space, while we do not have assumptions on the data distribution. (Senen-Cerda & Sanders, 2020) is closely related to our work. They studies the convergence properties of dropout stochastic gradient methods for minimizing the empirical risks of multiple fully connected networks. Different from (Li et al., 2016; Mianjy & Arora, 2020; Senen-Cerda & Sanders, 2020; Senen-Cerda & Sanders, 2022), we focus on bilevel training tasks.

Popular methods to solve bilevel optimization problems in machine learning have been proposed in (Franceschi et al., 2017; 2018; Finn et al., 2017a; Li et al., 2022; Gould et al., 2016; Lorraine et al., 2020a; Bae & Grosse, 2020). The major of them are gradient-based methods. They can be further divided into two types based on the way they approximate the hypergradient. The first type is iterative differentiation (ITD) (Franceschi et al., 2017; 2018; Finn et al., 2017a; Liu et al., 2020; Ghadimi & Wang, 2018; Ji et al., 2021; Rajeswaran et al., 2019), and the second type is approximate implicit differentiation (AID) (Chen et al., 2022; Ji et al., 2021; Li et al., 2022; Gould et al., 2016; Lorraine et al., 2020a). As far as we know, there is no existing work in bilevel optimization that considers the overfitting problem, let alone analyzing the dropout bilevel method for this problem. In this work, we select an ITD method with a relatively simple structure as an example to investigate how dropout affects bilevel training tasks.

## 2 PRELIMINARIES

In this paper, we denote $\mathbb{R}^n$ the $n$-dimensional Euclidean space with inner product $\langle \cdot, \cdot \rangle$ and Euclidean norm $\| \cdot \|$. We denote the spectrum norm of a matrix $A \in \mathbb{R}^{n \times m}$ as $\|A\|$ and the Frobenius norm of $A$ as $\|A\|_F$. For any matrices $A$ and $B$, we denote $\text{trace}(A^T B) := \langle A, B \rangle$. For

a ramdom variable $\xi$ defined on a probability space $(\Xi, \Sigma, P)$, we denote its expectation as $\mathbb{E}\xi$. Given an event $A$ and a function $f$ of $\xi$, the conditional expectation of $\xi$ is denoted as $\mathbb{E}_{\xi|A} f(\xi)$. For a function $F : \mathbb{R}^n \to \mathbb{R}^m$, we denote the function $F(x, y)$ with respect to $y$ for a fixed $x$ as $F(x, \cdot)$ and denote the function $F(x, y)$ with respect to $x$ for a fixed $y$ as $F(\cdot, y)$. For a differentiable function $f$, we say $x$ is an $\epsilon$ stationary point of $f$ if $\|\nabla f(x)\| \leq \epsilon$. For a twice differential function $F : \mathbb{R}^m \times \mathbb{R}^n \to \mathbb{R}$, we denote $\nabla_x F(x, y)$ and $\nabla_y F(x, y)$ as the partial gradients $\frac{\partial F(x,y)}{\partial y}$ and $\frac{\partial F(x,y)}{\partial y}$ correspondingly. We denote $\nabla^2 F(x, y)$ as the Hessian matrix of $F$, $\nabla_{xy} F(x, y) := \frac{\partial^2 F(x,y)}{\partial x \partial y}$ and $\nabla_{yy} F(x, y) := \frac{\partial^2 F(x,y)}{\partial y \partial y}$. For a multiple valued differentiable function $f : \mathbb{R}^n \to \mathbb{R}^m$, we denote its Jacobian at $x$ as $J(f(x))$. With a little abuse of notation, given a distribution $P$, let $g(x; \xi)$ be a function that depends on $x \in \mathbb{R}^n$ and a data point $\xi \sim P$.

In general, a bilevel optimization training task is formed as follows:

$$\min_{\lambda \in \mathbb{R}^{n_\lambda}} F(\lambda, w(\lambda)) := F(\lambda, w(\lambda); \mathcal{D}_{val}), \text{ s.t. } w(\lambda) \in \arg\min_{w \in \mathbb{R}^{n_w}} G(\lambda, w) := G(\lambda, w; \mathcal{D}_{tr}), \quad (3)$$

where $\mathcal{D}_{val}$ is a validation data set, $\mathcal{D}_{tr}$ is a training data set, $F : \mathbb{R}^{n_\lambda} \to \mathbb{R}$ and $G : \mathbb{R}^{n_\lambda + n_w} \to \mathbb{R}$ are differentiable functions[1].

As mentioned in Section 2, there are various approaches to solve (3). Due to the nested structure of bilevel problems, the methods for (3) are inherently complex. To better understand how dropout is implemented in methods for (3), we use the fully single loop algorithm (FSLA) proposed in (Li et al., 2022) as an example to investigate dropout. This method has relatively simple formulas at each iteration and low computational cost per iteration. Step 6 in FSLA is one step of SGD for the lower level problem and Step 9 generates an approximation of the hypergradient of the objective in the upper level.

## 2.1 DROPOUT

Let $f(w; \xi)$ be an $l$-layer network, where $w$ is the weight and $\xi$ is an input data. In particular, we assume

$$f(w; \xi) := f(w; \xi) = a_l(w_l(a_{l-1}(w_{l-1} \cdots a_1(w_1 \xi)))), \quad (4)$$

where $w_i \in \mathbb{R}^{n_i} \times \mathbb{R}^{o_i}$ is the parameters in the $i$th layer[2], $a_i$ is the activation function in the $i$th layer. We let $w = ((w_1)_1, \ldots, (w_1)_{n_1}, \ldots, (w_l)_1, \ldots, (w_l)_{n_l})$ be the collection of all rows in all weight matrices, where $(w_i)_s$ is the transpose of the $j$th row of $w_i$. Thus $w$ is an $no$-dimensional vector, where $n = \sum n_i$, $o = \sum o_i$. Let $l$ be a loss function. When training $f$ with loss function $l$, the forward pass is

$$F(w; \xi) = l(f(w; \xi); \xi). \quad (5)$$

In the $i$th layer, the forward pass of an input $\xi_i \in \mathbb{R}^{n_i}$ using dropout can be formed as

$$r_i = m_i \circ a_i(w_i \xi_i), \quad (6)$$

where $\xi_0 = \xi$, $\circ$ represent the Hadamard product of two vectors, and $m_i \in \mathbb{R}^{n_i}$ with $(m_i)_j \sim Bernoulli(p_i)$ with $p_i \in [0, 1]$. $m_j$ is called a dropout mask at the $i$th layer. Note that this formation of dropout masks is more general than the original dropout masks proposed in Srivastava et al. (2014). In Srivastava et al. (2014), the dropout rate is the same for all rows of weights in the same layer. Here, we allow the drop rate of each row of the weight matrix to be different.

Note that the function values of many well known activation functions at 0 is 0. For example, tanh, centered sigmoid and Relu has this property. Through out this work, we make the following assumption.

**Assumption 1.** *For any $a_i$ used in* (4)*, $a_i(0) = 0$.*

Under assumption 1, $r_i$ in (6) can be alternatively formed as

$$r_i = a_i(m_i \circ (w_i \xi_i)) = a_i \left( \begin{bmatrix} (m_i)_1 & 0 & 0 \\ \vdots & \ddots & \vdots \\ 0 & \ldots & (m_i)_{n_i} \end{bmatrix} w_i \xi_i \right). \quad (7)$$

---

[1] With a little abuse of notation, without misunderstanding, we denote the $F(\lambda, w(\lambda); \mathcal{D}_{val})$ as a function $F(\lambda)$ and denote $G(\lambda, w; \mathcal{D}_{tr})$ as a function $G(\lambda, w)$.

[2] For simplicity, we include the parameters of bias in $w_i$ with a corresponding fixed input of 1.

Denote $diag((m_i)_1, \ldots, (m_i)_{n_i}) := \begin{bmatrix} (m_i)_1 & 0 & 0 \\ \vdots & \ddots & \vdots \\ 0 & \ldots & (m_i)_{n_i} \end{bmatrix}$. Recalling that for any layer $i$, $(m_i)_j \sim Bernoulli(p_i)$. The right hand formula in (7) means the dropout method randomly pick the $j$th row of the weight matrix $w_i$ with probability $p_i$ and sets the rest to 0. Thus, given a loss function $l$, under Assumption 1, the forward pass with dropout in fact calculates

$$l(f(mw; \xi); \xi) = F(mw; \xi),$$

where $F$ is defined in (5) and

$$m := diag\left((m_1)_1, \ldots, (m_1)_{n_1}, \ldots, (m_l)_1, \ldots, (m_l)_{n_l}\right). \tag{8}$$

As we can see, the $j$th diagonal element $m_{jj} \sim Bernoulli(p_i)$ when $j \in \{\sum_k^{i-1} n_k + 1, \ldots, \sum_k^i n_k\}$. Note that in above discussion, each row of the weight matrix in each layer is always viewed as one element. For notation simplicity, without loss of generality, in the rest of this paper, when we mention the weight of a network, we view the weight matrix in the $i$th layer as a vector of $\mathbb{R}^{n_i}$, where $n_i$ is the number of outputs in the $i$th layer. Then, we view $w$, the collection of all rows of all layers, as a vector in $\mathbb{R}^n$.

## 3 Dropout in Bilevel Problems

As introduced in Section 2.1, the objectives are the composition of them with the dropout masks. Thus, we propose the following variant of (3) that considers the distribution of the dropout masks:

$$\min_{\lambda \in \mathbb{R}^{n_\lambda}} \tilde{F}(\lambda) := \mathbb{E}_{m_\lambda, m_w} F(m_\lambda \lambda, m_w w(\lambda)),$$
$$\text{s.t. } w(\lambda) \in \arg\min_{w \in \mathbb{R}^{n_w}} \tilde{G}(\lambda, w) := \mathbb{E}_{m_\lambda, m_w} G(m_\lambda \lambda, m_w w), \tag{9}$$

where $F$ and $G$ are the same as in (3), $m_\lambda$ and $m_w$ are random diagonal matrices with $(m_\lambda)_{ii} \sim Bernoulli(p_{i,\lambda})$, $(m_w)_{ii} \sim Bernoulli(p_{i,w})$ respectively. This idea of modeling with the dropout masks is also considered in analyzing dropout SGD for the single level training tasks in (Senen-Cerda & Sanders, 2020). For bilevel learning tasks, this model is new. We add dropout masks on both upper and lower level objective functions to include the application where $\lambda$ and $w$ are both weights of a network, Franceschi et al. (2018); Zügner & Günnemann (2019); Finn et al. (2017a); Snell et al. (2017).

Now we adopt the dropout bilevel method. based on FSLA. Before presenting a general form of FSLA with dropout, let's consider applying it to the example mentioned in (1). The data cleaning problem (1) is a case of (3) with $F(\lambda, w(\lambda); \mathcal{D}_{val}) = \frac{1}{|N_{val}|} \sum_{i=1}^{N_{val}} l(f(w(\lambda); \eta_i); \eta_i)$ and $G(\lambda, w; \mathcal{D}_{tr}) = \frac{1}{|N_{tr}|} \sum_{i=1}^{N_{tr}} \sigma(\lambda_i) l(f(w; \xi_i); \xi)$. As we illustrate in Section 2.1, suppose $m$ is a dropout masks added in the forward pass of the network used in (1) in each iteration. Then the forward pass of the upper level objective used in FSLA becomes $F(\lambda, mw(\lambda); \mathcal{D}_{val})$, where $m$ is defined in (8). The forward pass of the lower level objective used in FSLA becomes $G(\lambda, mw; \mathcal{D}_{tr})$. By the chain rule, the backward pass needed in FSLA calculate the follows:

$$\nabla F((\cdot), mw(\cdot); \mathcal{D}_{val})(\lambda) = m \nabla F((\cdot), mw(\cdot); \mathcal{D}_{val})(\lambda);$$
$$\nabla G(\cdot, m(\cdot); \mathcal{D}_{tr})(\lambda, w) = m \nabla G(\lambda, mw; \mathcal{D}_{tr}); \tag{10}$$
$$\nabla^2 G(\cdot, m(\cdot); \mathcal{D}_{tr})(\lambda, w) = m \nabla^2 G(\lambda, mw; \mathcal{D}_{tr}) m.$$

Based on this, we present Algorithm 1. We add a projection in step 8 to avoid $v$ blow up. This is also important in the theoretical analysis. In the next section, we analyze the convergence of Algorithm 1.

## 4 Analysis of Algorithm 1

A challenge of analyzing (9), which the single level model considered in (Senen-Cerda & Sanders, 2020) does not have, is how to analyze the hypergradient $\nabla \tilde{F}(\lambda)$. $\nabla \tilde{F}(\lambda)$ is hard to analyze due to

---

**Algorithm 1** FSLA with dropout

---

1: Input: $\beta, \alpha, \gamma > 0$, $\lambda^0$, $w^0$, dropout rates $\{p_{i,\lambda}\}_{i=1}^{n_\lambda}$, $\{p_{i,w}\}_{i=1}^{n_w}$, $r > 0$, $\nabla \bar{F}(\lambda^0, w^0, v^{-1}))$.
2: **for** $k = 0, \dots, T$. **do**
3:    $\alpha_k = \frac{\delta}{\sqrt{k}}$; $\gamma_k = \gamma \alpha_k$; $\beta_k = \beta \alpha_k$; $\eta_k = \eta \alpha_k$.
4:    Generate random diagonal matrices $\overline{m}_\lambda^k$, $\overline{m}_w^k$, $\underline{m}_\lambda^k$ and $\underline{m}_w^k$ with $(\overline{m}_\lambda^k)_{ii}, (\underline{m}_\lambda^k)_{ii} \sim$ $Bernoulli(p_{i,\lambda})$, $(\overline{m}_w^k)_{ii}, (\underline{m}_w^k)_{ii} \sim Bernoulli(p_{i,w})$.
5:    Sample $\xi^k \in \mathcal{D}_{tr}$, $\eta^k \in \mathcal{D}_{val}$.
6:    $w^{k+1} = w^k - \gamma_k \underline{m}_w^k \nabla_w G(\underline{m}_\lambda^k \lambda^k, \underline{m}_w^k w^k; \xi^k)$.
7:    $\tilde{v}^{k+1} = \beta \overline{m}_w^k \nabla_w F(\overline{m}_\lambda^k \lambda^k, \overline{m}_w^k w^{k+1}; \eta^k) + (I - \beta_k \underline{m}_w^k \nabla_{ww}^2 G(\underline{m}_\lambda^k \lambda^k, \underline{m}_w^k w^k; \xi^k) \underline{m}_w^k) v^k$.
8:    $v^{k+1} = \text{Proj}_{B(0,r)} \tilde{v}^{k+1}$.
9:    $\nabla \bar{F}(\lambda^k, w^{k+1}, v^{k+1}) = \overline{m}_\lambda^k \nabla_\lambda F(\overline{m}_\lambda^k \lambda^k, \overline{m}_w w^{k+1}; \eta^k) - \underline{m}_\lambda^k \nabla_{\lambda w}^2 G(\underline{m}_\lambda^k \lambda^k, \underline{m}_w^k w^{k+1}; \xi^k) \underline{m}_w^k v^{k+1}$.

10:    $d^{k+1} = \nabla \bar{F}(\lambda^k, w^{k+1}, v^{k+1}) + (1 - \eta_k)(d^k - \nabla \bar{F}(\lambda^{k-1}, w^k, v^k)))$.
11:    $\lambda^{k+1} = \lambda^k - \alpha_k d^{k+1}$.
12: **end for**

---

its relation with the solution of the lower level problem and the fact that both upper and lower level objectives are composed with random dropout masks. To analyze this and Algorithm 1 for (9), we first make the following assumptions that are standard in bilevel optimization literature.

**Assumption 2.** *Let $F(\lambda, w)$ be defined as in (3). Suppose the following assumptions hold:*

(i) *Suppose $F$ is Lipschitz continuous with modulus $L_F$.*

(ii) *For any fixed $\lambda$, $\nabla_\lambda F(\lambda, \cdot)$ and $\nabla_w F(w, \cdot)$ are Lipschitz continuous with modulus $L_{12}^F$ and $L_{22}^F$.*

(iii) *There exists $C^F > 0$ such that $\max\{\|\nabla_w F(\lambda, w)\|, \|\nabla_\lambda F(\lambda, w)\|\} \leq C^F$ for any $\lambda$ and $w$.*

(iv) *For any fixed $w$, $\nabla_w F(\cdot, w)$ is Lipschitz continuous with modulus $L_{w\lambda}^F > 0$.*

**Assumption 3.** *Let $G(\lambda, w)$ be defined as in (3). Suppose the following assumptions hold:*

(i) *$G$ is twice continuously differentiable. $\nabla G(\lambda, w)$ is Lipschitz continuous with modulus $L_G$.*

(ii) *For any $\lambda$, $G(\lambda, \cdot)$ is strongly convex with modulus $\mu$.*

(iii) *For any $\lambda$, $\nabla_{ij}^2 G(\lambda, \cdot)$ is Lipschitz continuous with modulus $L_{G^w}$.*

(iv) *For any $w, \nabla_{i,j}^2 G(\cdot, w)$ is Lipschitz continuous with modulus $L_{G^\lambda}$.*

(v) *There exists $C_G$ such that $\max_{i,j}\{\|\nabla_{ij}^2 G(\lambda, w)\|, \|\nabla_{ij}^2 G(\lambda, w)\|\} \leq C_G$ for any $\lambda$ and $w$.*

Denote $M = \begin{bmatrix} m_\lambda & 0 \\ 0 & m_w \end{bmatrix}$ and $F_M(\lambda) := F(m_\lambda(\cdot), m_w w(\cdot))(\lambda)$. Then

$$\nabla \tilde{F}(\lambda) = \mathbb{E}_{m_\lambda, m_w} \nabla F_M(\lambda). \tag{11}$$

Under Assumptions 2 and 3, it is easy to see that using the chain rule,

$$\nabla F_M(\lambda) = m_\lambda \nabla_\lambda F(m_\lambda \lambda, m_w w(\lambda)) + J(w(\lambda))^T m_w \nabla_w F(m_\lambda \lambda, m_w w(\lambda)), \tag{12}$$

where $J(w(\lambda))$ is the Jacobian of $w(\lambda)$. The next proposition gives a closed form of $J(w(\lambda))$ and its property.

**Proposition 1.** *Consider (9) and suppose Assumption 3 holds. Denote $W = \begin{bmatrix} \lambda \\ w \end{bmatrix}$ and $M = \begin{bmatrix} m_\lambda & 0 \\ 0 & m_w \end{bmatrix}$. Let $\underline{p}_w = \min_i p_{i,w}$. Then $\tilde{G}$ is strongly convex with modulus $\underline{p}_w \mu$ and the following equalities hold:*

$$\nabla \tilde{G}(\lambda, w) = \mathbb{E} M \nabla G(MW); \ \nabla^2 \tilde{G}(\lambda, w) = \mathbb{E} M \nabla^2 G(MW) M;$$

$$J(w(\lambda)) = - \left(\nabla^2_{ww}\tilde{G}(\lambda, w(\lambda))\right)^{-1} \nabla^2_{\lambda w}\tilde{G}(\lambda, w(\lambda)). \tag{13}$$

*In addition, for any $\lambda$, it holds that $w(\lambda)$ is Lipschitz continuous with some modulus $L_w$. Also, for any $w$, it holds that $\|J(\lambda(w))\|_F \leq \frac{C_G}{\mu}$.*

Based on Proposition 1, we can estimate the variance of $\nabla\tilde{G}(\lambda, w)$.

**Lemma 1.** *Let $G$ be defined as in (3). Let $m_\lambda$ and $m_w$ are diagonal matrices with $(m_\lambda)_{ii} \sim Bernoulli(p_{\lambda,i})$ and $(m_w)_{jj} \sim Bernoulli(p_{w,j})$. Let $p_i = p_{i,\lambda}$ if $i \in \{1, \dots, n_\lambda\}$ and $p_i = p_{i,w}$ if $i \in \{n_\lambda + 1, \dots, n_\lambda + n_w\}$. Then*

$$\mathbb{E}\|\nabla^2 G\left(M(\cdot)\right)(W) - \mathbb{E}_M \nabla^2 G\left(M(\cdot)\right)(W)\|^2 \leq \sum_{i,j} p_i p_j \left(1 - p_i p_j\right) C_G^2. \tag{14}$$

Next, we show how the update of the lower level parameters behaves. Since Algorithm 1 uses the stochastic gradient and Hessian. We add the following assumptions.

**Assumption 4.** *Consider (9). Let $\xi$ be randomly picked from $\mathcal{D}_{tr}$ and $\eta$ be randomly picked from $\mathcal{D}_{val}$. Suppose $\mathbb{E}_\eta \nabla_\lambda F(\lambda, w; \eta) = \nabla_\lambda F(\lambda, w)$ and $\mathbb{E}_\xi \nabla_w G(\lambda, w; \xi) = \nabla_w G(\lambda, w)$. Suppose $\mathbb{E}_\eta \|\nabla_\lambda F(\lambda, w; \eta) - \nabla_\lambda F(\lambda, w)\|^2 \leq \sigma^2$, $\mathbb{E}_\xi \|\nabla_w G(\lambda, w; \xi) - \nabla_w G(\lambda, w)\|^2 \leq \sigma^2$, and for any $i, j$, $\mathbb{E}_\xi \|\nabla^2_{i,j} G(\lambda, w; \xi) - \nabla^2_{i,j} G(\lambda, w)\| \leq \sigma_h^2$.*

Now we show the inductive relations of $\{w^k\}$ generated by Algorithm 1.

**Lemma 2.** *Consider (9). Suppose Assumptions 2, 3 and 4 hold. Let $\{(\lambda^k, w^k)\}$ be generated by Algorithm 1. Let $\gamma$ in Algorithm 1 be small enough such that $\gamma\mu < 1$. Let $\overline{p}_w := \max_i p_{i,w}$. Then there exists $\iota \in (0, 1)$ such that*

$$\mathbb{E}\|w^{k+1} - w(\lambda^k)\|^2 \leq \iota \mathbb{E}\|w^k - w(\lambda^{k-1})\|^2 + O(\alpha_k^2)\mathbb{E}\|d^{k-1}\|^2 + O(\gamma_k^2)\overline{p}_w \sigma^2. \tag{15}$$

**Remark 1.** *Note that $w(\lambda^k)$ is the true solution of the lower level problem given $\lambda^k$. This induction has additional errors related to the updates of the upper level variable $\lambda^k$ and the variance. Note that the variance term in (15) decreases linearly with the maximum dropout rate.*

Next, we present the inductive relations of $\{v^k\}$ generated by Algorithm 1. In fact, combining (13) with (12), it is easy to see that

$$\nabla F_M(\lambda) = m_\lambda \nabla_\lambda F(m_\lambda \lambda, m_w w(\lambda)) - \nabla^2_{\lambda w}\tilde{G}(\lambda, w(\lambda))v_{m_w,\lambda}. \tag{16}$$

where $v_{m_w,\lambda} := \left(\nabla^2_{ww}\tilde{G}(\lambda, w(\lambda))\right)^{-1} m_w \nabla_w F(m_\lambda \lambda, m_w w(\lambda))$. This implies that

$$v_{m_w,\lambda} = \beta m_w \nabla_w F(m_\lambda \lambda, m_w w(\lambda)) + \left(I - \beta\nabla^2_{ww}\tilde{G}(\lambda, w(\lambda))\right) v_{m_w,\lambda}.$$

Comparing the above equation with the update of $v^{k+1}$ in Algorithm 1, we see that $v^{k+1}$ is an approximation of $v_{\overline{m}_w^k, \lambda^k}$. The next theorem shows how the difference $v^k$ and $v_{\overline{m}_w^k, \lambda^k}$ varies with iteration $k$ and the dropout rate.

**Theorem 1.** *Consider (9). Suppose Assumptions 2, 3 and 4 hold. Let $\{(\lambda^k, w^k)\}$ be generated by Algorithm 1. Denote $V^k := v^k - v_{\overline{m}_w^{k-1}, \lambda^{k-1}}$. Denote $\overline{p}_\lambda := \max_i p_{i,\lambda}$, $\underline{p}_\lambda := \min_i p_{i,\lambda}$, $\overline{p}_w := \max_i p_{i,w}$ and $\underline{p}_w := \min_i p_{i,w}$. Suppose $r \geq C_v := \frac{L^F_{22}}{\mu}$. Then there exists $\varrho \in (0, 1)$ such that*

$$\mathbb{E}\|V^{k+1}\|^2 \leq \varrho\mathbb{E}\|V^k\|^2 + O((\overline{p}_w)^2\beta_k^2)\sigma^2 + O(\overline{p}_w\beta_k^2)\mathbb{E}\|w^{k+1} - w(\lambda^k)\|^2$$
$$+ O(\alpha_k^2)\mathbb{E}\|d^{k-1}\|^2 + O((\overline{p}_w)^2\beta_k^2)\sigma_h^2 + O(\overline{p}_\lambda\overline{p}_w(1 - \underline{p}_\lambda\underline{p}_w)\beta_k^2).$$

**Remark 2.** *The last term in the above theorem are introduced by the dropout of the network and disappear when no dropout is applied to the network parametrized by $w$.*

Now, based on Theorem 1 and Lemma 2, we analyze the convergence of Algorithm 1. Thanks to (11), to find an $\epsilon$ stationary point to (9) such that $\|\nabla\tilde{F}(\lambda)\|^2 \leq \epsilon$, it suffices to find $\lambda$ that satisfies $\mathbb{E}\|\nabla F_M(\lambda)\|^2 \leq \epsilon$.

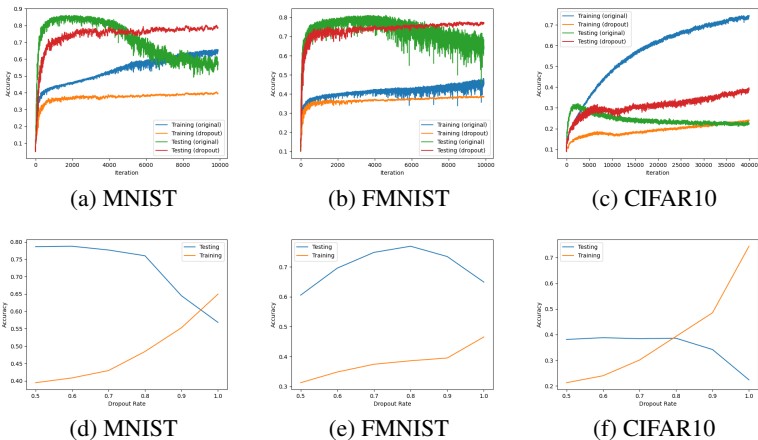

Figure 2: Results from data cleaning. The first line reports how training and testing accuracy change as the number of iterations increases. The second line details the training and testing accuracy in the final iteration when different dropout rates are applied.

**Theorem 2.** *Consider* (9). *Suppose Assumptions 2, 3 and 4 hold. Let* $\{(w^k, \lambda^k)\}$ *be generated by Algorithm 1. Suppose $F$ is bounded below by $\bar{F}$. Then there exist small $\delta$ in Algorithm 1 such that*

$$\frac{1}{T-1}\sum_{k=1}^{T-1}\mathbb{E}\|\nabla F_{M^k}(\lambda^k)\|^2 \leq \frac{1}{\sqrt{T}}\frac{1}{\underline{p}_w\underline{p}_\lambda}I_0 + \frac{\ln(T+1)}{\sqrt{T}}O\left(\frac{\bar{p}_\lambda^2}{\underline{p}_\lambda\underline{p}_w} + \underline{p}_w\bar{p}_w\right)(\sigma^2 + \sigma_h^2)$$
$$+ \frac{\ln(T+1)}{\sqrt{T}}O\left(\underline{p}_w\left(\bar{p}_w^2(1-\underline{p}_w^2) + \bar{p}_w(1-\underline{p}_w)\right)\right).$$

**Remark 3.** *The first term shows that the convergence rate is slower when $\underline{p}_\lambda\underline{p}_w$ is small, which can be confirmed in our experiments Figure 2. The last term in the above results is introduced by dropout, and it disappears when $\underline{p}_\lambda = \underline{p}_w = 1$, i.e., no dropout is applied.*

*On the other hand, we have to point that the convergence of $\mathbb{E}\nabla F_{M^k}(\lambda^k)$ does not imply that of $F(\lambda)$. Consider a simple 2-dimensional case. Let $f(x_1, x_2) := \frac{1}{2}x_1^2 x_2 + 1000x_1$. Let $x^k = (x_1^k, x_2^k)$ be with $x_1^k = -\frac{1000p^2 + 2000(1-p)p}{k}$ and $x_2^k = k$. Let $m = \text{diag}\{m_1, m_2\}$ with $m_1 \approx Bernoulli(p)$ and $m_2 \approx Bernoulli(p)$. Then $\lim_{k\to\infty}\mathbb{E}_m\nabla f(mx^k) = 0$. However, $\lim_{k\to\infty}\nabla f(x^k) = (-1000p^2 - 2000(1-p)p + 1000, 0)$. Therefore, the convergence of $\nabla\mathbb{E}_m f(mx^k)$ does not necessarily imply the convergence of $\nabla f(x^k)$, the error between $\nabla f(x^k)$ and $\nabla\mathbb{E}_m f(mx^k)$ can not be closed unless $p$ approaches 1. Therefore, there are additional errors brought only by the dropout. This example implies that the dropout method may not be optimizing the original bilevel problem. This is expected because when the original training loss is optimized, the model fits the training data too well and this will increase the chance of overfitting.*

## 5 EXPERIMENTS

In this section, we test Algorithm 1 on data cleaning tasks (1) and meta learning problem (2). The experiments were conducted on a machine equipped with an Intel Xeon E5-2683 CPU and 4 Nvidia Tesla P40 GPUs.

**Data cleaning** We followed the approach outlined in (Srivastava et al., 2014) and utilized a fully connected network. The experiments were performed using the MNIST dataset (LeCun et al., 2010). The network architecture employed was 784-1024-1024-2048-10, with ReLU activation functions used for all hidden units. To introduce unclean data, we randomly mislabel $60\%$ of the training data. When training on MNIST and FMNIST, we use a fully connected network with the architecture $784 - 1024 - 1024 - 2048 - 10$ and employ ReLU activation functions for all hidden units. In all experiments conducted on MNIST and FMNIST, we set the dropout rates of all layers to the same value, denoted as $p$. Dropout is only applied when updating the lower level parameters. We set $\gamma = 0.01$ and train $10000/10000$ iterations for $5000/10000$ MNIST/FMNIST data points. When

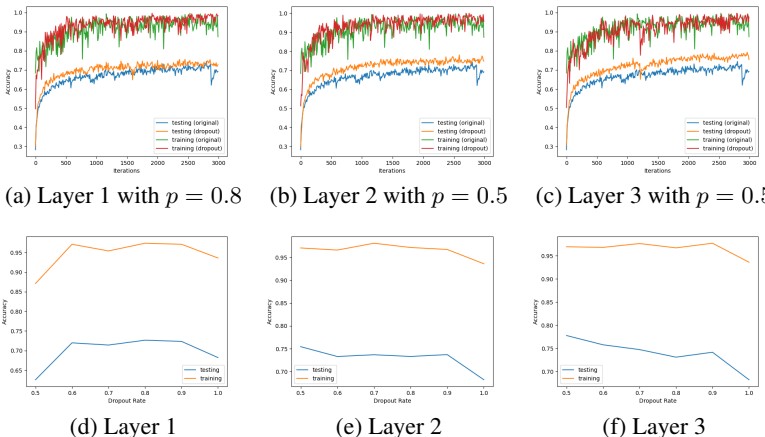

(a) Layer 1 with $p = 0.8$    (b) Layer 2 with $p = 0.5$    (c) Layer 3 with $p = 0.5$

(d) Layer 1              (e) Layer 2              (f) Layer 3

Figure 3: Results from few-shot learning on the Omniglot dataset. The first line reports how training and testing accuracy change as the number of iterations increases. The second line details the training and testing accuracy in the final iteration when different dropout rates are applied to only one layer of the network.

training on CIFAR10, we use 4-layer convolutional neural networks + 1 fully connected layer. In all CIFAR10 experiments, we apply dropout after each convolutional layer, using a consistent dropout rate of $p$. We set $\gamma = 0.1$. We train $40000$ iterations for $40000$ data points.

**Meta-learning** We conduct experiments with the few-shot learning task, following the experimental protocols of (Vinyals et al., 2016), we performed learning tasks over the Omniglot dataset. We set train/validation/test with 102/172/423, respectively. We perform 5-way-1-shot classification. More specifically, we perform 5 training tasks ($N = 5$). For each task, we randomly sample 5 characters from the alphabet over that client and for each character, and select 1 data points for training and 15 samples for validation. We use a 4-layer convolutional neural network + 1 fully connected layer. Each convolutional layer has 64 filters of 3×3 and is followed by batch-normalization layers (Finn et al., 2017b). The parameters of convolutional layers are shared between different tasks ($\lambda$) and the last linear layer is the task-specific parameter $w_j$. In each experiment, we only add dropout to one CNN layer. In all experiments, we let $\beta$, $\alpha$ and $\gamma$ in FSLA and Algorithm 1 be $0.05$, $0.1$ and $0.8$ respectively.

**Results** We report the results in Figures 2 and 3. In the first line of both figures, we plot the accuracy against the iteration in each training progress. In the second line of both figures, we plot how different dropout rates affect the training and testing accuracy. Figure 2 shows when the training accuracy increases, the testing accuracy decreases. This observation indicates the occurrence of overfitting in data cleaning tasks. The proper addition of dropout during training can enhance testing accuracy in response. On the other hand, Figure 2 demonstrates that training the network with dropout leads to convergence at a higher testing accuracy and greater stability. Figure 3 shows the accuracy of the method when adding dropout with different rates on the different layers. As we can see, adding a proper dropout to any layer improves the testing accuracy.

## 6  CONCLUSION

In this paper, we explore the application of dropout in bilevel training tasks. We propose a stochastic bilevel model that is dependent on the distribution of dropout masks. We adapt an existing bilevel method with dropout. We analyze the convergence properties of the resulting method for the proposed model. We investigate the inductive relations of the variables attributes to an approximation of the hypergradients. In addition, we show how the dropout rates affect the complexity. We believe that other state-of-art bilevel methods can also be adapted to address the stochastic bilevel model with random dropout masks, and our convergence analysis serve as the first example for such adaptations.

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
