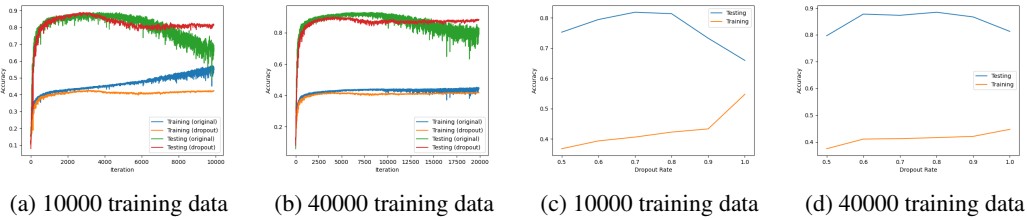

(a) 10000 training data    (b) 40000 training data    (c) 10000 training data    (d) 40000 training data

Figure 4: MNIST

Amirreza Shaban, Ching-An Cheng, Nathan Hatch, and Byron Boots. Truncated back-propagation for bilevel optimization. In *The 22nd International Conference on Artificial Intelligence and Statistics, AISTATS 2019, 16-18 April, Naha, Okinawa, Japan*, 2019.

Jake Snell, Kevin Swersky, and Richard S. Zemel. Prototypical networks for few-shot learning. In *Advances in Neural Information Processing Systems 30: Annual Conference on Neural Information Processing Systems 2017, December 4-9, Long Beach, CA, USA*, 2017.

Nitish Srivastava, Geoffrey E. Hinton, Alex Krizhevsky, Ilya Sutskever, and Ruslan Salakhutdinov. Dropout: a simple way to prevent neural networks from overfitting. *J. Mach. Learn. Res.*, 15(1): 1929–1958, 2014.

Oriol Vinyals, Charles Blundell, Tim Lillicrap, Koray Kavukcuoglu, and Daan Wierstra. Matching networks for one shot learning. In *Advances in Neural Information Processing Systems 29: Annual Conference on Neural Information Processing Systems 2016, December 5-10, Barcelona, Spain*, 2016.

Haotian Wang, Wenjing Yang, Zhenyu Zhao, Tingjin Luo, Ji Wang, and Yuhua Tang. Rademacher dropout: An adaptive dropout for deep neural network via optimizing generalization gap. *Neurocomputing*, 357:177–187, 2019. doi: 10.1016/j.neucom.2019.05.008. URL https://doi.org/10.1016/j.neucom.2019.05.008.

Haibing Wu and Xiaodong Gu. Towards dropout training for convolutional neural networks. *Neural Networks*, 71:1–10, 2015. ISSN 0893-6080. doi: https://doi.org/10.1016/j.neunet.2015.07.007. URL https://www.sciencedirect.com/science/article/pii/S0893608015001446.

Zhuoran Yang, Yongxin Chen, Mingyi Hong, and Zhaoran Wang. Provably global convergence of actor-critic: A case for linear quadratic regulator with ergodic cost. In *Advances in Neural Information Processing Systems*, volume 32. Curran Associates, Inc., 2019. URL https://proceedings.neurips.cc/paper_files/paper/2019/file/9713faa264b94e2bf346a1bb52587fd8-Paper.pdf.

Wojciech Zaremba, Ilya Sutskever, and Oriol Vinyals. Recurrent neural network regularization. *CoRR*, abs/1409.2329, 2014. URL http://arxiv.org/abs/1409.2329.

Ke Zhai and Huan Wang. Adaptive dropout with rademacher complexity regularization. In *International Conference on Learning Representations*, 2018. URL https://openreview.net/forum?id=S1uxsye0Z.

Daniel Zügner and Stephan Günnemann. Adversarial attacks on graph neural networks via meta learning. In *7th International Conference on Learning Representations, ICLR 2019, New Orleans, LA, USA, May 6-9*, 2019.

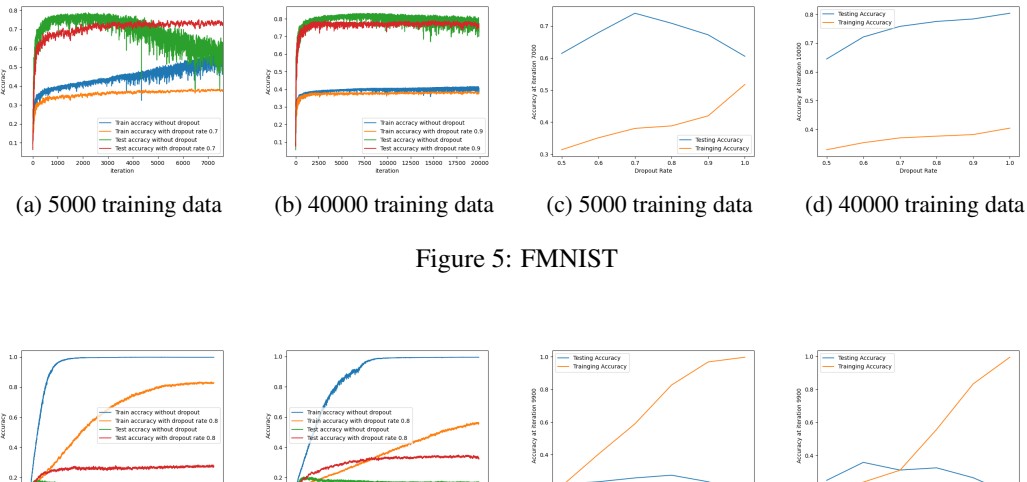

|  |  |  |  |
|---|---|---|---|
| (a) 5000 training data | (b) 40000 training data | (c) 5000 training data | (d) 40000 training data |

Figure 5: FMNIST

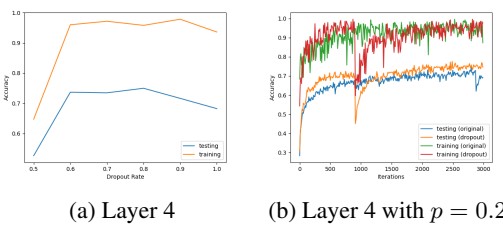

|  |  |  |  |
|---|---|---|---|
| (a) 5000 training data | (b) 10000 training data | (c) 5000 training data | (d) 10000 training data |

Figure 6: CIFAR10

|  |  |
|---|---|
| (a) Layer 4 | (b) Layer 4 with $p = 0.2$ |

Figure 7: Mata-learning

# A  ADDITIONAL EXPERIMENTS

## A.1  DATA CLEANING

We use the same network and hyperparameters of the FLSA with dropout as in the full paper. We train $10000/20000$ iterations for $10000/40000$ MNIST. We train $10000/20000$ iterations for $5000/40000$ for FMNIST. The results are in Figure 4 and 5.

Figure 2d 4c 4d indicates that as the dataset size increases, the lines representing testing accuracy and training accuracy become more parallel. This implies that in larger datasets, overfitting is less likely to happen.

In data learning rasks on CIFAR10, we let each convolutional layer be defined by its input channel, output channel, kernel size, stride, and padding, set as [64, 3, 3, 3, 2, 0], [64, 64, 3, 3, 2, 0], [64, 64, 3, 3, 2, 0], and [4, 64, 2, 2, 1, 0], respectively. The linear layer connects an input of size 64*4 to an output size of 10. Throughout the network, ReLU activation functions are employed for all hidden units. Additionally, we train $10000/10000$ iterations for $5000/10000$ data points. The results are in 6.

## A.2  META-LEARNING

We additionally test how dropout affects the last convolution layer, see Figure 7.

## B  PROOFS OF PROPOSITION 1

We restate a detailed version of the property here.

**Proposition 2.** *Consider* (9) *and suppose Assumption 3 holds. Denote* $W = \begin{bmatrix} \lambda \\ w \end{bmatrix}$ *and* $M = \begin{bmatrix} m_\lambda & 0 \\ 0 & m_w \end{bmatrix}$. *Let* $\underline{p}_w = \min_i p_{i,w}$. *Then* $\tilde{G}$ *is strongly convex with modulus* $\underline{p}_w \mu$ *and*

$$\nabla \tilde{G}(\lambda, w) = \sum_{m_w \in M_w, m_\lambda \in M_\lambda} p_{m_w, m_\lambda} M \nabla G(MW) = \mathbb{E} M \nabla G(MW);$$

$$\nabla^2 \tilde{G}(\lambda, w) = \mathbb{E} M \nabla^2 G(MW) M. \tag{17}$$

*Also* $w(\lambda)$ *is Lipschitz continuous with constant* $L_w := \frac{C_g}{\underline{p}_w \mu}$. *In addition, for any* $\lambda$, *it holds that* $J(w(\lambda)) = -\left( \nabla^2_{ww} \tilde{G}(\lambda, w(\lambda)) \right)^{-1} \nabla^2_{\lambda w} \tilde{G}(\lambda, w(\lambda))$ *is Lipschitz continuous with modulus* $L_J := \frac{L_{G_\lambda^w} + L_{G_\lambda^w} L_w}{\underline{p}_w \mu} + \frac{(L_{G_w^\lambda} + L_{G_w^w} L_w) C_G}{\underline{p}_w^2 \mu^2}$. *Also, for any* $w$, *it holds that* $\|J(\lambda(w))\|_F \leq \frac{C_G}{\underline{p}_w \mu}$.

*Proof.* Note that

$$\tilde{G}(\lambda, w) = \sum_{m_w \in M_w, m_\lambda \in M_\lambda} p_{m_w, m_\lambda} G(m_\lambda \lambda, m_w w)$$

where $P_{m_w, m_\lambda}$ is the probability of the case $(m_w, m_\lambda)$ happends and $(M_w, M_\lambda)$ is the set of all cases of $m_w, m_\lambda$. Thus $\tilde{G}(\lambda, w)$ is also twice continuously differentiable.

Note that $G(m_\lambda(\cdot), m_w(\cdot))(\lambda, m_w)$ can be equally written as $G\left( \begin{bmatrix} m_\lambda & 0 \\ 0 & m_w \end{bmatrix} (\cdot) \right) \left( \begin{bmatrix} \lambda \\ w \end{bmatrix} \right)$. Then,

$$\nabla \tilde{G}(\lambda, w) = \sum_{m_w \in M_w, m_\lambda \in M_\lambda} p_{m_w, m_\lambda} M \nabla G(MW) = \mathbb{E} M \nabla G(MW). \tag{18}$$

and

$$\nabla^2 \tilde{G}(\lambda, w) = \sum_{m_w \in M_w, m_\lambda \in M_\lambda} p_{m_w, m_\lambda} M \nabla^2 G(MW) M = \mathbb{E} M \nabla^2 G(MW) M. \tag{19}$$

These together with Assumption 3, the fact that $p_{m_w, m_\lambda} \in [0, 1]$ and the fact that all entries of the diagonal matrix $M$ belongs to $\{0, 1\}$ shows that

- $\tilde{G}$ is twice continuously differentiable. In addition, suppose $\nabla_w \tilde{G}(\lambda, \cdot)$ is Lipschitz continuous with modulus $L_G > 0$ for any $\lambda$.
- For any $\lambda$, $\nabla^2_{\lambda w} \tilde{G}(\lambda, \cdot)$ and $\nabla^2_{ww} \tilde{G}(\lambda, \cdot)$ are Lipschitz continuous with modulus $L_{G_\lambda^w}$ and $L_{G_w^w}$.

Now we show that for any $\lambda$, $\tilde{G}(\lambda, \cdot)$ is strongly convex with modulus $\mu$. Let $v \in \mathbb{R}^{n_w}$. Using (19), we have that

$$v \nabla^2_{ww} \tilde{G}(\lambda, w) v = v \left( \mathbb{E} m_w \nabla^2 G(m_\lambda \lambda, m_w w) m_w \right) v$$

$$= \mathbb{E} m_w v \nabla^2 G(m_\lambda \lambda, m_w w) m_w v$$

$$\geq \mathbb{E} \mu \| m_w v \|^2 = \sum_{i=1}^{n_w} \mathbb{E}_{(m_w)_{ii}} \mu \| (m_w)_{ii} v_i \|^2 \tag{20}$$

$$= \sum_{i=1}^{n_w} \mu p_{i,w} \| v_i \|^2 \geq \mu \underline{p}_w \| v \|^2,$$

where the first inequality uses Assumption 3 and $\underline{p}_w = \min_i p_{i,w}$. Thus, using Lemma 2.1 and Lemma 2.2 in (Ghadimi & Wang, 2018), for any $\lambda$, it holds that

- $w(\lambda)$ is Lipschitz continuous with constant $\frac{C_g}{\underline{p}_w \mu}$.

- 
$$J(w(\lambda)) = - \left( \nabla^2_{ww} \tilde{G}(\lambda, w(\lambda)) \right)^{-1} \nabla^2_{\lambda w} \tilde{G}(\lambda, w(\lambda)) \tag{21}$$

is Lipschitz continuous with modulus $L_J := \frac{L_{G^\lambda} + L_{G^w_\lambda} L_w}{\underline{p}_w \mu} + \frac{(L_{G^\lambda_w} + L_{G^w_w} L_w) C_G}{\underline{p}^2_w \mu^2}$. Also, for any $w$, it holds that $\|J(w(\lambda))\|_F \leq \frac{C_G}{\underline{p}_w \mu}$.

$\square$

## C  PROOFS OF LEMMA 1

We restate the lemma here.

**Lemma 3.** *Let $G$ be defined as in* (3)*. Let $m_\lambda$ and $m_w$ are diagonal matrices with $(m_\lambda)_{ii} \sim Bernoulli(p_{\lambda,i})$ and $(m_w)_{jj} \sim Bernoulli(p_{w,j})$. Let $p_i = p_{i,\lambda}$ if $i \in \{1, \ldots, n_\lambda\}$ and $p_i = p_{i,w}$ if $i \in \{n_\lambda + 1, \ldots, n_\lambda + n_w\}$. Then*

$$\mathbb{E}\|\nabla^2 G\left(M(\cdot)\right)(W) - \mathbb{E}_M \nabla^2 G\left(M(\cdot)\right)(W)\|^2 \leq \sum_{i,j} p_i p_j \left(1 - p_i p_j\right) C_G^2. \tag{22}$$

*Similarly, we have*

$$\mathbb{E}\|\nabla_w F\left(M(\cdot)\right)(W) - \mathbb{E}_M \nabla_w F\left(M(\cdot)\right)(W)\|^2 \leq \sum_{i,j} p_i \left(1 - p_i\right) \left(C^F\right)^2. \tag{23}$$

*Proof.* Note that $G(m_\lambda(\cdot), m_w(\cdot))(\lambda, m_w)$ can be equally written as $G\left( \begin{bmatrix} m_\lambda & 0 \\ 0 & \underline{m}_w \end{bmatrix} (\cdot) \right) \left( \begin{bmatrix} \lambda \\ w \end{bmatrix} \right)$.

Denote $W = \begin{bmatrix} \lambda \\ w \end{bmatrix}$ and $M = \begin{bmatrix} m_\lambda & 0 \\ 0 & \underline{m}_w \end{bmatrix}$. Then,

$$\nabla G\left(M(\cdot)\right)(W) = M \nabla G\left(MW\right)$$

and

$$\begin{aligned}
\nabla^2 G\left(M(\cdot)\right)(W) &= M \nabla^2 G\left(MW\right) M \\
&= \sum_{i,j} diag(0, \ldots, 0, M_{ii}, 0, \ldots, 0) H_G\left(MW\right) diag(0, \ldots, 0, M_{jj}, 0, \ldots, 0) \\
&= \sum_{i,j} \begin{bmatrix} 0 & 0 & & \cdots & & \cdots & 0 \\ \vdots & \vdots & & \vdots & & \vdots & \vdots \\ 0 & 0 & M_{ii} \frac{\partial^2 G}{\partial W_j \partial W_i}\left(MW\right) M_{jj} & \cdots & & & 0 \\ \vdots & \vdots & & \vdots & & \vdots & \vdots \\ 0 & 0 & & \cdots & & \cdots & 0 \end{bmatrix}
\end{aligned} \tag{24}$$

Thus,

$$\mathbb{E}_M \nabla^2 G\left(M(\cdot)\right)(W) = \mathbb{E}_M \sum_{i,j} \begin{bmatrix} 0 & 0 & & \cdots & & \cdots & 0 \\ \vdots & \vdots & & \vdots & & \vdots & \vdots \\ 0 & 0 & p_i p_j \frac{\partial^2 G}{\partial W_j \partial W_i}\left(MW\right) & \cdots & & & 0 \\ \vdots & \vdots & & \vdots & & \vdots & \vdots \\ 0 & 0 & & \cdots & & \cdots & 0 \end{bmatrix}$$

Fix any $i_0, j_0 \in \{1, \ldots, n_\lambda + n_w\}$. It holds that

$$
\mathbb{E}_M \left\| M_{i_0 i_0} H_G (MW) M_{j_0 j_0} - \mathbb{E}_M \frac{\partial^2 G (MW)}{\partial W_j \partial W_i} (W) \right\|^2
$$

$$
= \mathbb{E}_M \left\| M_{i_0 i_0} \frac{\partial^2 G(MW)}{\partial W_j \partial W_i} M_{j_0 j_0} - p_{i_0} p_{j_0} \frac{\partial^2 G (MW)}{\partial W_j \partial W_i} \right\|^2
$$

$$
= p_{i_0} p_{j_0} \left\| \frac{\partial^2 G (MW)}{\partial W_j \partial W_i} - p_{i_0} p_{j_0} \frac{\partial^2 G (MW)}{\partial W_j \partial W_i} \right\|^2 + (1 - p_{i_0} p_{j_0}) \left\| p_{i_0} p_{j_0} \frac{\partial^2 G}{\partial W_j \partial W_i} (MW) \right\|^2
$$

$$
\leq p_{i_0} p_{j_0} (1 - p_{i_0} p_{j_0})^2 \left\| \frac{\partial^2 G (MW)}{\partial W_j \partial W_i} \right\|^2 + (1 - p_{i_0} p_{j_0}) (p_{i_0} p_{j_0})^2 \left\| \frac{\partial^2 G}{\partial W_j \partial W_i} (MW) \right\|^2
$$

$$
\leq \left( p_{i_0} p_{j_0} (1 - p_{i_0} p_{j_0})^2 + (1 - p_{i_0} p_{j_0}) (p_{i_0} p_{j_0})^2 \right) C_G^2 = (p_{i_0} p_{j_0} (1 - p_{i_0} p_{j_0})) C_G^2.
$$

Since $i_0$ and $j_0$ are arbitrarily chosen, we have that

$$
\mathbb{E} \| \nabla^2 G (M(\cdot)) (W) - \mathbb{E}_M \nabla^2 G (M(\cdot)) (W) \|^2
$$
$$
\leq \mathbb{E} \| \nabla^2 G (M(\cdot)) (W) - \mathbb{E}_M \nabla^2 G (M(\cdot)) (W) \|_F^2 \leq \sum_{i_0, j_0} p_{i_0} p_{j_0} (1 - p_{i_0} p_{j_0}) C_G^2.
$$

$\square$

## D    PROOFS OF LEMMA 2

We first present a lemma that will be used repeatedly.

**Lemma 4.** *Let $m_1 \in \mathbb{R}^{n_1} \times \mathbb{R}^{n_1}$ and $m_2 \in \mathbb{R}^{n_2} \times \mathbb{R}^{n_2}$ be diagonal matrices. If $\|v(m_1)\|^2 \leq c_1$ for a constant $c_1$, then*
$$
\mathbb{E}_{m_1} \| m_1 v(m_1) \|^2 \leq \max_j p_j c_1.
$$

*If $\|A_{ij}\| \leq c$,*
$$
\mathbb{E}_{m_1, m_2} \| m_1 A m_2 \|^2 \leq \max p_j \max p_i n_1 n_2 c_2^2.
$$

*Proof.* Noting $\|m_1 v(m_1)\|^2 = \sum_j (m_1)_{jj} \|v_j(m)\|^2$, it holds that

$$
\mathbb{E}_{m_1} \| m_1 v \|^2 \leq \mathbb{E}_{m_1} \sum_j (m_1)_{jj} \|v(m_1)_j\|^2 = \sum_j p_j \mathbb{E}_{m_1 | (m_1)_{jj} = 1} \|v(m_1)_j\|^2
$$
$$
\leq \sum_j (\max_{j'} p_{j'}) \|v(m_1)_j\|^2 \leq \max_j p_j c_1.
$$

On the other hand,

$$
\mathbb{E}_{m_1, m_2} \| m_1 A m_2 \|^2 \leq \mathbb{E}_{m_1, m_2} \sum_{i,j} (m_1)_{jj} A_{ji}^2 (m_2)_{ii} \leq \max p_j \max p_i n_1 n_2 c_2^2.
$$

$\square$

We present a detailed version of Lemma 2 here.

**Lemma 5.** *Consider (9). Suppose Assumptions 2, 3 and 4 hold. Let $\{(\lambda^k, w^k)\}$ be generated by Algorithm 1. Let $\gamma$ in Algorithm 1 be small enough such that $\gamma \mu < 1$. Then*

$$
\mathbb{E} \| w^{k+1} - w(\lambda^k) \|^2 \leq (1 - \frac{1}{2} \gamma_k \underline{p}_w \mu) \mathbb{E} \| w^k - w(\lambda^{k-1}) \|^2
$$
$$
+ (\frac{2}{\gamma_k \underline{p}_w \mu} - 1)(1 - \gamma_k \underline{p}_w \mu) L_w^2 \alpha_k^2 \mathbb{E} \| d^{k-1} \|^2 + \gamma_k^2 \overline{p}_w \sigma^2
$$

(25)

*and*

$$\mathbb{E}\|w^{T+2} - w(\lambda^{T+1})\|^2 - \mathbb{E}\|w^1 - w(\lambda^0)\|^2 \leq -\frac{1}{2}\sum_{k=1}^{T+1}\gamma_k \underline{p}_w \mu \mathbb{E}\|w^k - w(\lambda^{k-1})\|^2$$
$$+ \sum_{k=1}^{T+1}(\frac{2}{\gamma_k \underline{p}_w \mu} - 1)(1 - \gamma_k \underline{p}_w \mu)L_w^2\alpha_k^2\mathbb{E}\|d^{k-1}\|^2 + \sum_{k=1}^{T+1}\gamma_k^2\overline{p}_w\sigma^2. \tag{26}$$

*Proof.* Using the Lipschitz continuity of $\tilde{G}$, it holds that Using the definition of $w^{k+1}$ in Algorithm 1,

$$\mathbb{E}_{\underline{m}_\lambda^k,\underline{m}_w^k,\xi^k|\mathcal{M}^{k-1}}\|w^{k+1} - w(\lambda^k)\|^2 = \|w^k - \gamma_k \underline{m}_w^k \nabla_w G(\underline{m}_\lambda^k\lambda^k,\underline{m}_w^k w^k;\xi^k) - w(\lambda^k)\|^2$$
$$= \|w^k - w(\lambda^k)\|^2 + \gamma_k\left\langle\nabla\tilde{G}(\lambda^k,w^k),w^k - w(\lambda^k)\right\rangle$$
$$+ \mathbb{E}_{\underline{m}_\lambda^k,\underline{m}_w^k,\xi^k|\mathcal{M}^{k-1}}\|\gamma_k\underline{m}_w^k\nabla_w G(\underline{m}_\lambda^k\lambda^k,\underline{m}_w^k w^k;\xi^k))\|^2 \tag{27}$$
$$\leq \|w^k - w(\lambda^k)\|^2 + \gamma_k\left\langle\nabla\tilde{G}(\lambda^k,w^k),w^k - w(\lambda^k)\right\rangle + \gamma_k^2\overline{p}_w\sigma^2,$$

where the first equality uses Assumption 4 and (17), the second inequality follows from Assumption 4 and Lemma 4. Note from Proposition 2 that $\tilde{G}$ is strongly convex and $w(\lambda^k)$ is the minimizer of $\min_w \tilde{G}(\lambda^k,w)$. Thus,

$$\left\langle\nabla\tilde{G}(\lambda^k,w^k),w^k - w(\lambda^k)\right\rangle \geq \underline{p}_w\mu\|w^k - w(\lambda^k)\|^2.$$

This together with (27) gives

$$\mathbb{E}_{\underline{m}_\lambda^k,\underline{m}_w^k,\xi^k|\mathcal{M}^{k-1}}\|w^{k+1} - w(\lambda^k)\|^2 \leq (1 - \gamma_k\underline{p}_w\mu)\|w^k - w(\lambda^k)\|^2 + \gamma_k^2\overline{p}_w\sigma^2$$
$$\leq (1 + \kappa_3^k)(1 - \gamma_k\underline{p}_w\mu)\|w^k - w(\lambda^{k-1})\|^2 + (1 + \frac{1}{\kappa_3^k})(1 - \gamma_k\underline{p}_w\mu)\|w(\lambda^k) - w(\lambda^{k-1})\|^2 + \gamma_k^2\overline{p}_w\sigma^2$$
$$\overset{(a)}{\leq} (1 + \kappa_3^k)(1 - \gamma_k\underline{p}_w\mu)\|w^k - w(\lambda^{k-1})\|^2 + (1 + \frac{1}{\kappa_3^k})(1 - \gamma_k\underline{p}_w\mu)L_w^2\alpha_k^2\|d^{k-1}\|^2 + \gamma_k^2\overline{p}_w\sigma^2$$
$$= (1 - \frac{1}{2}\gamma_k\underline{p}_w\mu)\|w^k - w(\lambda^{k-1})\|^2 + (\frac{2}{\gamma_k\underline{p}_w\mu} - 1)(1 - \gamma_k\underline{p}_w\mu)L_w^2\alpha_k^2\|d^{k-1}\|^2 + \gamma_k^2\overline{p}_w\sigma^2$$

where $\kappa_3^k := \frac{1}{2}\frac{\gamma_k\underline{p}_w\mu}{1-\gamma_k\underline{p}_w\mu}$ such that $(1 + \kappa_3^k)(1 - \gamma_k\underline{p}_w\mu) < 1$, (a) makes use the definition of $\lambda^k$ and the fact that $w(\lambda)$ is $L_w$-Lipschitz continuous by Proposition 2, and the last inequality makes use of the definition of $\kappa_3^k$. Taking expectation w.r.t. $\mathcal{M}^{k-1}$ we obtain

$$\mathbb{E}\|w^{k+1} - w(\lambda^k)\|^2$$
$$\leq (1 - \frac{1}{2}\gamma_k\underline{p}_w\mu)\mathbb{E}\|w^k - w(\lambda^{k-1})\|^2 + (\frac{2}{\gamma_k\underline{p}_w\mu} - 1)(1 - \gamma_k\underline{p}_w\mu)L_w^2\alpha_k^2\mathbb{E}\|d^{k-1}\|^2 + \gamma_k^2\overline{p}_w\sigma^2,$$

Rearranging the above inequality, we have

$$\mathbb{E}\|w^{k+1} - w(\lambda^k)\|^2 - \mathbb{E}\|w^k - w(\lambda^{k-1})\|^2 \leq -\frac{1}{2}\gamma_k\underline{p}_w\mu\mathbb{E}\|w^k - w(\lambda^{k-1})\|^2$$
$$+ (\frac{2}{\gamma_k\underline{p}_w\mu} - 1)(1 - \gamma_k\underline{p}_w\mu)L_w^2\alpha_k^2\mathbb{E}\|d^{k-1}\|^2 + \gamma_k^2\overline{p}_w\sigma^2.$$

Summing the above inequality from $k = 1$ to $k = T$ and divide both sides with $T$, we have

$$\mathbb{E}\|w^{T+1} - w(\lambda^T)\|^2 - \mathbb{E}\|w^1 - w(\lambda^0)\|^2 \leq -\frac{1}{2}\sum_{k=1}^{T}\gamma_k\underline{p}_w\mu\mathbb{E}\|w^k - w(\lambda^{k-1})\|^2$$
$$+ \sum_{k=1}^{T}(\frac{2}{\gamma_k\underline{p}_w\mu} - 1)(1 - \gamma_k\underline{p}_w\mu)L_w^2\alpha_k^2\mathbb{E}\|d^{k-1}\|^2 + \sum_{k=1}^{T}\gamma_k^2\overline{p}_w\sigma^2.$$

□

## E  PROOFS OF THEOREM 1

We first show the property of $v_{m,\lambda}$ w.r.t. $\lambda$.

**Lemma 6.** *Consider* (9). *Suppose Assumptions 2, 3 and 4 hold. Let* $\{(\lambda^k, w^k)\}$ *be generated by Algorithm 1. Let* $v_{\overline{m}_w^k, \lambda^k}$ *be defined in* (16). *Denote* $\overline{p}_\lambda := \max_i p_{i,\lambda}$, $\underline{p}_\lambda := \min_i p_{i,\lambda}$, $\overline{p}_w := \max_i p_{i,w}$ *and* $\underline{p}_w := \min_i p_{i,w}$. *Then the following conclusions hold:*

*(i) There exists $C_v$ such that $\|v_{\overline{m}_w^k, \lambda^k}\| \le C_v$.*

*(ii) There exists $C_w$ such that*

$$\|\mathbb{E}_{M^k} v_{\overline{m}_w^k, \lambda^k} - \mathbb{E}_{M^{k-1}} v_{\overline{m}_w^{k-1}, \lambda^{k-1}}\|^2 \le C_w \alpha_k^2 \|d^k\|^2.$$

*Proof.* (i) follows from Assumptions 2 and 3. Now we prove (ii). Denote $\mathcal{M}^k := \{M^1, \ldots, M^k, \xi_1, \ldots, \xi_k, \ldots, \eta_1, \ldots, \eta_k\}$. Denote $F_M(\lambda) := F(m_\lambda(\cdot), m_w w(\cdot))(\lambda)$. It holds that

$$
\begin{aligned}
&\mathbb{E}_{M^k} v_{\overline{m}_w^k, \lambda^k} - \mathbb{E}_{M^{k-1}} v_{\overline{m}_w^{k-1}, \lambda^{k-1}} \\
&= \left(\nabla_{ww}^2 \tilde{G}(\lambda^k, w(\lambda^k))\right)^{-1} \nabla_w F_{M^k}(\lambda^k, w(\lambda^k)) \\
&\quad - \left(\nabla_{ww}^2 \tilde{G}(\lambda^{k-1}, w(\lambda^{k-1}))\right)^{-1} \nabla_w F_{M^{k-1}}(\lambda^{k-1}, w(\lambda^{k-1})) \\
&= \left(\left(\nabla_{ww}^2 \tilde{G}(\lambda^k, w(\lambda^k))\right)^{-1} - \left(\nabla_{ww}^2 \tilde{G}(\lambda^{k-1}, w(\lambda^{k-1}))\right)^{-1}\right) \nabla_w F_{M^k}(\lambda^k, w(\lambda^k)) \\
&\quad + \left(\nabla_{ww}^2 \tilde{G}(\lambda^{k-1}, w(\lambda^{k-1}))\right)^{-1} \left(\nabla_w F_{M^k}(\lambda, w(\lambda^k)) - \nabla_w F_{M^{k-1}}(\lambda^{k-1}, w(\lambda^{k-1}))\right).
\end{aligned}
\tag{28}
$$

Thanks to Assumption 3, there exists $L_{G^{-1}}$ such that

$$\left\|\left(\nabla_{ww}^2 \tilde{G}(\lambda^k, w(\lambda^k))\right)^{-1} - \left(\nabla_{ww}^2 \tilde{G}(\lambda^{k-1}, w(\lambda^{k-1}))\right)^{-1}\right\|^2 \le L_{G^{-1}}^2 \|w(\lambda^k) - w(\lambda^{k-1})\|^2. \tag{29}$$

Combining this with (28), (29) with Assumptions 2 and 3, we have that

$$
\begin{aligned}
&\|\mathbb{E}_{M^k} v_{\overline{m}_w^k, \lambda^k} - \mathbb{E}_{M^{k-1}} v_{\overline{m}_w^{k-1}, \lambda^{k-1}}\|^2 \\
&\le L_{G^{-1}}^2 C^F \|w(\lambda^k) - w(\lambda^{k-1})\|^2 + \frac{1}{\underline{p}_w^2 \mu^2} \overline{p}_w^2 (L_{ww}^F)^2 \left(\alpha_k^2 \|d^k\|^2 + \|w(\lambda^k) - w(\lambda^{k-1})\|^2\right) \\
&\le \left(L_{G^{-1}}^2 C^F \left(\frac{C_g}{\underline{p}_w \mu}\right)^2 + \frac{1}{\underline{p}_w^2 \mu^2} \overline{p}_w^2 (L_{ww}^F)^2 (1 + \left(\frac{C_g}{\underline{p}_w \mu}\right)^2)\right) \alpha_k^2 \|d^k\|^2.
\end{aligned}
$$

where the second inequality uses (4) and the third inquelity uses Proposition 2. $\qquad\square$

Now we present the detailed version of Theorem 1.

**Theorem 3.** *Consider* (9). *Suppose Assumptions 2, 3 and 4 hold. Let* $\{(\lambda^k, w^k)\}$ *be generated by Algorithm 1. Denote* $V^k := \mathbb{E}\|v^k - \mathbb{E}_{M^{k-1}} v_{\overline{m}_w^{k-1}, \lambda^{k-1}})\|^2$, *where* $v_{\overline{m}_w^{k-1}, \lambda^{k-1}}$ *is defined in* (16). *Denote* $\overline{p}_\lambda := \max_i p_{i,\lambda}$, $\underline{p}_\lambda := \min_i p_{i,\lambda}$, $\overline{p}_w := \max_i p_{i,w}$ *and* $\underline{p}_w := \min_i p_{i,w}$. *Suppose* $r \ge C_v$, *where $C_v$ is defined as in Lemma 6 (i). Then*

$$
\begin{aligned}
V^{k+1} &\le \frac{2}{\underline{p}_w \mu} \beta_k L_{ww}^F \overline{p}_w \mathbb{E}\|w^{k+1} - w(\lambda^k)\|^2 + (1 - \frac{1}{2}\beta_k \underline{p}_w \mu)^2 V^k + \frac{2}{\beta \underline{p}_w \mu} C_w \alpha_k \mathbb{E}\|d^k\|^2 \\
&\quad + \beta_k^2 \left(4\overline{p}_w \sigma^2 + 4\overline{p}_w^2 \sigma_h^2 + 4\overline{p}_w^2 (1 - \underline{p}_w^2) n C_G^2 + 4\overline{p}_w (1 - \underline{p}_w) n (C^F)^2\right).
\end{aligned}
$$

*and*

$$V^{T+2} - V^1 \leq \sum_{k=1}^{T+1} \left( -\frac{1}{2}\underline{p}_w\mu + \frac{1}{4}\beta_k\underline{p}_w^2\mu^2 \right) \beta_k V^k$$

$$+ \frac{2}{\underline{p}_w\mu} \sum_{k=1}^{T+1} \beta_k L_{ww}^F \bar{p}_w \mathbb{E}\|w^{k+1} - w(\lambda^k)\|^2 + \frac{2}{\beta\underline{p}_w\mu} C_w \sum_{k=1}^{T+1} \alpha_k \mathbb{E}\|d^k\|^2 \tag{30}$$

$$+ \sum_{k=1}^{T+1} \beta_k^2 \left( 4\bar{p}_w\sigma^2 + 4\bar{p}_w^2\sigma_h^2 + 4\bar{p}_w^2(1-\underline{p}_w^2)nC_G^2 + 4\bar{p}_w(1-\underline{p}_w)n(C^F)^2 \right).$$

*Proof.* Denote $\mathcal{M}^k := \{M^1, \ldots, M^k, \xi_1, \ldots, \xi_k, \ldots, \eta_1, \ldots, \eta_k\}$. Denote $\mathcal{B}^k := \{\xi^k, \eta^k\}$. Denote $F_M(\lambda) := F(m_\lambda(\cdot), m_w w(\cdot))(\lambda)$.

Let $\tilde{v}_d^{k+1} := \beta_k \nabla_w \mathbb{E}_{M^k} F_{M^k}(\lambda^k, w^{k+1}) - \left( I - \beta_k \nabla_{ww}^2 \tilde{G}(\lambda^k, w^k) \right) v^k$. Therefore, $\mathbb{E}_{M^k,\xi^k,\eta^k} v_d^{k+1} = \tilde{v}_d^{k+1}$. Using Assumption 2, Assumption 3, Lemma 3 and Lemma 4, we have

$$v_d^{k+1} \mathbb{E}_{M^k,\xi^k,\eta^k} \|\tilde{v}_d^{k+1} - \mathbb{E}_{M^k} v_{\overline{m}_w^k, \lambda^k}\|^2$$
$$\leq \beta_k^2 \left( 4\bar{p}_w\sigma^2 + 4\bar{p}_w^2\sigma_h^2 + 4\bar{p}_w^2(1-\underline{p}_w^2)nC_G^2 + 4\bar{p}_w(1-\underline{p}_w)n(C^F)^2 \right). \tag{31}$$

On the other hand, the definition of $v_{\overline{m}_w^k, \lambda^k}$ shows that

$$\mathbb{E}_{M^k} v_{\overline{m}_w^k, \lambda^k} = \beta_k \nabla_w \mathbb{E}_{M^k} F_{M^k}(\lambda^k, w(\lambda^k)) + \left( I - \beta_k \nabla_{ww}^2 \tilde{G}(\lambda^k, w(\lambda^k)) \right) \mathbb{E}_M^k v_{\overline{m}_w^k, \lambda^k}.$$

Combining this with (31) and the definition of $v^{k+1}$, we have that

$$\mathbb{E}\|v^{k+1} - \mathbb{E}_{M^k} v_{\overline{m}_w^k, \lambda^k}\|^2 \leq \|\tilde{v}^{k+1} - v_{\overline{m}_w^k, \lambda^k}\|^2$$
$$= \mathbb{E}\|\tilde{v}^{k+1} - v_{\overline{m}_w^k, \lambda^k} - \tilde{v}_d^{k+1} - v_{\overline{m}_w^k, \lambda^k} + \tilde{v}_d^{k+1} - v_{\overline{m}_w^k, \lambda^k}\|^2$$
$$= \mathbb{E}\|\tilde{v}^{k+1} - \tilde{v}_d^{k+1}\|^2 + \|\tilde{v}_d^{k+1} - v_{\overline{m}_w^k, \lambda^k}\|^2$$
$$\leq (1 + \frac{1}{\kappa_1^k})\beta_k^2 \mathbb{E}\|\mathbb{E}_{M^k} F_{M^k}(\lambda^k, w^k) - \mathbb{E}_{M^k} F_{M^k}(\lambda^k, w(\lambda^k))\|^2 \tag{32}$$
$$+ (1 + \kappa_1^k)\mathbb{E}\| \left( I - \beta_k \nabla_{ww}^2 \tilde{G}(\lambda^k, w(\lambda^k)) \right) (v^k - \mathbb{E}_{M^k} v_{\overline{m}_w^k, \lambda^k})\|^2$$
$$+ \beta_k^2 \left( 4\bar{p}_w\sigma^2 + 4\bar{p}_w^2\sigma_h^2 + 4\bar{p}_w^2(1-\underline{p}_w^2)nC_G^2 + 4\bar{p}_w(1-\underline{p}_w)n(C^F)^2 \right),$$

where the first inequality is because $r \geq C_v$ and the projection on $B(0, r)$ is a contraction.

For the first term in the above inequality, using Lemma 4 and Assumption 2, it holds that

$$\mathbb{E}\|\mathbb{E}_{M^k} F_{M^k}(\lambda^k, w^k) - \mathbb{E}_{M^k} F_{M^k}(\lambda^k, w(\lambda^k))\|^2 \leq L_{ww}^F \bar{p}_w \|w^{k+1} - w(\lambda^k)\|^2. \tag{33}$$

For the second term in (32), using Proposition 2, it holds that

$$\| \left( I - \beta_k \nabla_{ww}^2 \tilde{G}(\lambda^k, w(\lambda^k)) \right) \mathbb{E}_{M^k}(v^k - v_{\overline{m}_w^k, \lambda^k})\|^2$$
$$\leq (1 - \beta_k\underline{p}_w\mu)^2 \|(v^k - \mathbb{E}_{M^k} v_{\overline{m}_w^k, \lambda^k})\|^2$$
$$\leq (1 + \kappa_2^k)(1 - \beta_k\underline{p}_w\mu)^2 \|(v^k - \mathbb{E}_{M^{k-1}} v_{\overline{m}_w^{k-1}, \lambda^{k-1}})\|^2 \tag{34}$$
$$+ (1 + \frac{1}{k_2^k})(1 - \beta_k\underline{p}_w\mu)^2 \|\mathbb{E}_{M^{k-1}} v_{\overline{m}_w^{k-1}, \lambda^{k-1}} - \mathbb{E}_{M^k} v_{\overline{m}_w^k, \lambda^k}\|^2.$$

where $\kappa_2^k > 0$. This together with Lemma 6 implies that

$$\| \left( I - \beta_k \nabla_{ww}^2 \tilde{G}(\lambda^k, w(\lambda^k)) \right) \mathbb{E}_{M^k}(v^k - v_{\overline{m}_w^k, \lambda^k})\|^2$$
$$\leq (1 - \beta_k\underline{p}_w\mu)^2 \|(v^k - \mathbb{E}_{M^k} v_{\overline{m}_w^k, \lambda^k})\|^2$$
$$\leq (1 + \kappa_2^k)(1 - \beta_k\underline{p}_w\mu)^2 \|(v^k - \mathbb{E}_{M^{k-1}} v_{\overline{m}_w^{k-1}, \lambda^{k-1}})\|^2 + (1 + \frac{1}{k_2^k})(1 - \beta_k\underline{p}_w\mu)^2 C_w\alpha_k^2\|d^k\|^2$$
$$\tag{35}$$

Combining (35), (33) and (32), we have that

$$
\begin{aligned}
\mathbb{E}\|v^{k+1} - \mathbb{E}_{M^k} v_{\overline{m}_w^k, \lambda^k}\|^2 &\leq \|\tilde{v}^{k+1} - v_{\overline{m}_w^k, \lambda^k}\|^2 \\
&= \mathbb{E}\|\tilde{v}^{k+1} - v_{\overline{m}_w^k, \lambda^k} - \tilde{v}_d^{k+1} - v_{\overline{m}_w^k, \lambda^k} + \tilde{v}_d^{k+1} - v_{\overline{m}_w^k, \lambda^k}\|^2 \\
&= \mathbb{E}\|\tilde{v}^{k+1} - \tilde{v}_d^{k+1}\|^2 + \|\tilde{v}_d^{k+1} - v_{\overline{m}_w^k, \lambda^k}\|^2 \\
&\leq (1 + \frac{1}{\kappa_1^k})\beta_k^2 L_{ww}^F \overline{p}_w \|w^{k+1} - w(\lambda^k)\|^2 \\
&\quad + (1 + \kappa_1^k)(1 + \kappa_2^k)(1 - \beta_k \underline{p}_w \mu)^2 \|(v^k - \mathbb{E}_{M^{k-1}} v_{\overline{m}_w^{k-1}, \lambda^{k-1}})\|^2 \\
&\quad + (1 + \kappa_1^k)(1 + \frac{1}{k_2^k})(1 - \beta_k \underline{p}_w \mu)^2 C_w \alpha_k^2 \|d^k\|^2 \\
&\quad + \beta_k^2 \left(4\bar{p}_w \sigma^2 + 4\bar{p}_w^2 \sigma_h^2 + 4\bar{p}_w^2(1 - \underline{p}_w^2)nC_G^2 + 4\bar{p}_w(1 - \underline{p}_w)n(C^F)^2\right).
\end{aligned}
\tag{36}
$$

Let $\kappa_1 = \kappa_2 = \frac{\beta_k \underline{p}_w \mu}{2(1 - \beta_k \underline{p}_w \mu)}$. Then $1 + \kappa_1 = \frac{1}{2} + \frac{1}{2 - \beta_k \underline{p}_w \mu}$, $1 + \frac{1}{\kappa_1} = 1 + \frac{1}{\kappa_2} = \frac{2 - \beta_k \underline{p}_w \mu}{\beta_k \underline{p}_w \mu} \leq \frac{2}{\beta_k \underline{p}_w \mu}$.
Thus, (36)

$$
\begin{aligned}
\mathbb{E}\|v^{k+1} - \mathbb{E}_{M^k} v_{\overline{m}_w^k, \lambda^k}\|^2 &\leq \|\tilde{v}^{k+1} - v_{\overline{m}_w^k, \lambda^k}\|^2 \\
&\leq \frac{2}{\beta_k \underline{p}_w \mu} \beta_k^2 L_{ww}^F \overline{p}_w \|w^{k+1} - w(\lambda^k)\|^2 \\
&\quad + (\frac{1}{2} + \frac{1}{2 - \beta_k \underline{p}_w \mu})(\frac{1}{2} + \frac{1}{2 - \beta_k \underline{p}_w \mu})(1 - \beta_k \underline{p}_w \mu)^2 \|(v^k - \mathbb{E}_{M^{k-1}} v_{\overline{m}_w^{k-1}, \lambda^{k-1}})\|^2 \\
&\quad + (\frac{1}{2} + \frac{1}{2 - \beta_k \underline{p}_w \mu})\frac{2}{\beta_k \underline{p}_w \mu}(1 - \beta_k \underline{p}_w \mu)^2 C_w \alpha_k^2 \|d^k\|^2 \\
&\quad + \beta_k^2 \left(4\bar{p}_w \sigma^2 + 4\bar{p}_w^2 \sigma_h^2 + 4\bar{p}_w^2(1 - \underline{p}_w^2)nC_G^2 + 4\bar{p}_w(1 - \underline{p}_w)n(C^F)^2\right) \\
&\leq \frac{2}{\underline{p}_w \mu} \beta_k L_{ww}^F \overline{p}_w \mathbb{E}\|w^{k+1} - w(\lambda^k)\|^2 \\
&\quad + (1 - \frac{1}{2}\beta_k \underline{p}_w \mu)^2 \mathbb{E}\|v^k - \mathbb{E}_{M^{k-1}} v_{\overline{m}_w^{k-1}, \lambda^{k-1}}\|^2 + \frac{2}{\beta \underline{p}_w \mu} C_w \alpha_k \mathbb{E}\|d^k\|^2 \\
&\quad + \beta_k^2 \left(4\bar{p}_w \sigma^2 + 4\bar{p}_w^2 \sigma_h^2 + 4\bar{p}_w^2(1 - \underline{p}_w^2)nC_G^2 + 4\bar{p}_w(1 - \underline{p}_w)n(C^F)^2\right).
\end{aligned}
$$

Rearranging the above inequality, we have

$$
\begin{aligned}
\mathbb{E}\|v^{k+1} - \mathbb{E}_{M^k} v_{\overline{m}_w^k, \lambda^k}\|^2 &- \mathbb{E}\|v^k - \mathbb{E}_{M^{k-1}} v_{\overline{m}_w^{k-1}, \lambda^{k-1}}\|^2 \\
&\leq \left(-\frac{1}{2}\underline{p}_w \mu + \frac{1}{4}\beta_k \underline{p}_w^2 \mu^2\right) \beta_k \mathbb{E}\|v^k - \mathbb{E}_{M^{k-1}} v_{\overline{m}_w^{k-1}, \lambda^{k-1}}\|^2 \\
&\quad + \frac{2}{\underline{p}_w \mu} \beta_k L_{ww}^F \overline{p}_w \mathbb{E}\|w^{k+1} - w(\lambda^k)\|^2 + \frac{2}{\beta \underline{p}_w \mu} C_w \alpha_k \mathbb{E}\|d^k\|^2 \\
&\quad + \beta_k^2 \left(4\bar{p}_w \sigma^2 + 4\bar{p}_w^2 \sigma_h^2 + 4\bar{p}_w^2(1 - \underline{p}_w^2)nC_G^2 + 4\bar{p}_w(1 - \underline{p}_w)n(C^F)^2\right).
\end{aligned}
$$

Summing the above inequality from $k = 1$ to $k = T + 1$, we have

$$
\begin{aligned}
\mathbb{E}\|v^{T+2} - \mathbb{E}_{M^{T+1}} v_{\overline{m}_w^{T+1}, \lambda^{T+1}}\|^2 &- \mathbb{E}\|v^1 - \mathbb{E}_{M^0} v_{\overline{m}_w^0, \lambda^0}\|^2 \\
&\leq \sum_{k=1}^{T+1} \left(-\frac{1}{2}\underline{p}_w \mu + \frac{1}{4}\beta_k \underline{p}_w^2 \mu^2\right) \beta_k \mathbb{E}\|v^k - \mathbb{E}_{M^{k-1}} v_{\overline{m}_w^{k-1}, \lambda^{k-1}}\|^2 \\
&\quad + \frac{2}{\underline{p}_w \mu} \sum_{k=1}^{T+1} \beta_k L_{ww}^F \overline{p}_w \mathbb{E}\|w^{k+1} - w(\lambda^k)\|^2 + \frac{2}{\beta \underline{p}_w \mu} C_w \sum_{k=1}^{T+1} \alpha_k \mathbb{E}\|d^k\|^2 \\
&\quad + \sum_{k=1}^{T+1} \beta_k^2 \left(4\bar{p}_w \sigma^2 + 4\bar{p}_w^2 \sigma_h^2 + 4\bar{p}_w^2(1 - \underline{p}_w^2)nC_G^2 + 4\bar{p}_w(1 - \underline{p}_w)n(C^F)^2\right).
\end{aligned}
$$

$\square$

## F    PROOF OF THEOREM 2

We first present the following lemma that is needed in the proofs of Theorem 2.

**Lemma 7.** *Consider* (9)*. Let assumptions in Theorem 2 hold. Then, it holds that*

$$
\mathbb{E}F_{M^{k+1}}(\lambda^{k+1}) - \mathbb{E}F_{M^k}(\lambda^k) + \Gamma_3 \left( \mathbb{E}\|d^{k+1} - \bar{\nabla}\bar{F}(\lambda^k, w^{k+1}, v^{k+1})\|^2 - \mathbb{E}\|(d^k - \bar{\nabla}\bar{F}(\lambda^{k-1}, w^k, v^k)\|^2 \right)
$$

$$
\leq -\frac{\alpha_k}{2}\mathbb{E}\|\nabla F_{M^k}(\lambda^k)\|^2
$$

$$
- \Gamma_3 \alpha_k \eta_k \mathbb{E}\|(d^k - \bar{\nabla}\bar{F}(\lambda^{k-1}, w^k, v^k)\|^2 + \Gamma_3 \alpha_k \eta_k^2 \left( (\bar{p}_\lambda)^2 \sigma^2 + \underline{p}_\lambda \underline{p}_w \sigma_h^2 \right)
$$

$$
+ \Gamma_3 \alpha_k \bar{p}_\lambda \left( 2(L_{12}^F)^2 + 2r^2(1 + 3L_w^2) + 2C_G^2 3 C_w \right) \alpha_{k-1}^2 \mathbb{E}\|d^{k-1}\|^2
$$

$$
+ \Gamma_3 \alpha_k \underline{p}_\lambda \left( 2(L_{12}^F)^2 + 2r^2 \right) \left( 3\mathbb{E}\|w^k - w(\lambda^{k-1})\|^2 + 3\mathbb{E}\|w^{k+1} - w(\lambda^k)\|^2 \right)
$$

$$
+ 2\Gamma_3 \alpha_k \underline{p}_\lambda \underline{p}_w C_G^2 \left( 3V^k + 3V^{k+1} \right)
$$

$$
+ \alpha_k \bar{p}_w \left( (L_{12}^F)^2 + nL_{G^w}^2 r^2 \right) \mathbb{E}\|w^{k+1} - w(\lambda^k)\|^2 + \alpha_k \bar{p}_\lambda \bar{p}_w n C_G^2 V^{k+1}
$$

$$
+ \left( \frac{\alpha_k^2 L_{\bar{F}}}{2} - \frac{\alpha_k}{2} \right) \mathbb{E}\|d^{k+1}\|^2,
$$

$$(37)$$

*Proof.* Denote $\mathcal{M}^k := \{M^1, \ldots, M^k, \xi_1, \ldots, \xi_k, \ldots, \eta_1, \ldots, \eta_k\}$. Denote $\mathcal{B}^k := \{\xi^k, \eta^k\}$. Denote $F_M(\lambda) := F(m_\lambda(\cdot), m_w w(\cdot))(\lambda)$.

Thanks to Proposition 2, there exists $L_{\bar{F}}$ such that $\nabla F_M(\lambda)$ is $L_{\bar{F}}$-Lipschitz continuous. Thus,

$$
\mathbb{E}F_{M^{k+1}}(\lambda^{k+1}) \leq \mathbb{E}F_{M^{k+1}}(\lambda^k) + \mathbb{E}\left\langle \nabla F_{M^{k+1}}(\lambda^k), \lambda^{k+1} - \lambda^k \right\rangle + \mathbb{E}\frac{L_{\bar{F}}}{2}\|\lambda^{k+1} - \lambda^k\|^2
$$

$$
= \mathbb{E}F_{M^k}(\lambda^k) - \alpha_k \mathbb{E}\left\langle \nabla F_{M^k}(\lambda^k), d^{k+1} \right\rangle + \alpha_k^2 \mathbb{E}\frac{L_{\bar{F}}}{2}\|d^{k+1}\|^2
$$

$$
= \mathbb{E}F_{M^k}(\lambda^k) - \frac{\alpha_k}{2}\mathbb{E}\|\nabla F_{M^k}(\lambda^k)\|^2 + \frac{\alpha_k}{2}\mathbb{E}\|\nabla F_{M^k}(\lambda^k) - d^{k+1}\|^2
$$

$$
+ \left( \frac{\alpha_k^2 L_{\bar{F}}}{2} - \frac{\alpha_k}{2} \right) \mathbb{E}\|d^{k+1}\|^2,
$$

$$(38)$$

where the first equality is because $M^{k+1}$ and $M^k$ have the same distributions and therefore

$$
\mathbb{E}F_{M^{k+1}}(\lambda^k) = \mathbb{E}_{\mathcal{M}^{k-1}}\mathbb{E}_{M^{k+1},M^k|\mathcal{M}^{k-1}}F_{M^{k+1}}(\lambda^k)
$$

$$
= \mathbb{E}_{\mathcal{M}^{k-1}}\mathbb{E}_{M^k|\mathcal{M}^{k-1}}F_{M^k}(\lambda^k) = \mathbb{E}F_{M^k}(\lambda^k).
$$

and

$$
\mathbb{E}\nabla F_{M^{k+1}}(\lambda^k) = \mathbb{E}_{\mathcal{M}^{k-1}}\mathbb{E}_{M^{k+1},M^k|\mathcal{M}^{k-1}}\nabla F_{M^{k+1}}(\lambda^k)
$$

$$
= \mathbb{E}_{\mathcal{M}^{k-1}}\mathbb{E}_{M^k|\mathcal{M}^{k-1}}\nabla F_{M^k}(\lambda^k) = \mathbb{E}\nabla F_{M^k}(\lambda^k).
$$

Next, we bound the third term on the right hand side of (38). Using the chain rule, it holds that

$$
\nabla F_M(\lambda) = m_\lambda \nabla_\lambda F(m_\lambda \lambda, m_w w(\lambda)) + J(w(\lambda))^T m_w \nabla_w F(m_\lambda \lambda, m_w w(\lambda)),
$$

where $J(w(\lambda))$ is the Jacobian of $w(\lambda)$. Using Proposition 2, the above equality ca be further passed to

$$
\nabla F_M(\lambda)
$$

$$
= m_\lambda \nabla_\lambda F(m_\lambda \lambda, m_w w(\lambda)) - \nabla_{\lambda w}^2 \tilde{G}(\lambda, w(\lambda)) \underbrace{\left( \nabla_{ww}^2 \tilde{G}(\lambda, w(\lambda)) \right)^{-1} m_w \nabla_w F(m_\lambda \lambda, m_w w(\lambda))}_{v_{m_w, \lambda}}.
$$

Denote

$$
\bar{\nabla}\bar{F}(\lambda^k, w^{k+1}, v^{k+1}) := \bar{m}_\lambda^k \nabla_\lambda F(\bar{m}_\lambda^k \lambda^k, \bar{m}_w w^{k+1}) - \underline{m}_{\lambda^k} \nabla_{\lambda w}^2 G(\underline{m}_\lambda^k \lambda^k, \underline{m}_w^k w^{k+1})\underline{m}_w^k v^{k+1}.
$$

$$(39)$$

This together with the definition of $d^{k+1}$ in Algorithm 1 gives

$$
\begin{aligned}
&\|d^{k+1} - \nabla F_{M^k}(\lambda^k)\|^2 \\
&\leq 2\|d^{k+1} - \bar{\nabla}\bar{F}(\lambda^k, w^{k+1}, v^{k+1})\|^2 + 2\|\bar{\nabla}\bar{F}(\lambda^k, w^{k+1}, v^{k+1}) - \nabla F_{M^k}(\lambda^k)\|^2.
\end{aligned}
\tag{40}
$$

Now we bound the first term on the right hand side of the above inequality.

$$
\begin{aligned}
&\mathbb{E}\|d^{k+1} - \bar{\nabla}\bar{F}(\lambda^k, w^{k+1}, v^{k+1})\|^2 \\
&= \mathbb{E}\|\nabla\bar{F}(\lambda^k, w^{k+1}) + (1-\eta_k)(d^k - \nabla\bar{F}(\lambda^{k-1}, w^k))) - \bar{\nabla}\bar{F}(\lambda^k, w^{k+1}, v^{k+1})\|^2 \\
&= (1-\eta_k)^2 \mathbb{E}\|d^k - \bar{\nabla}\bar{F}(\lambda^{k-1}, w^k, v^k)\|^2 \\
&\quad + \mathbb{E}\|(1-\eta_k)(\bar{\nabla}\bar{F}(\lambda^{k-1}, w^k, v^k) - \nabla\bar{F}(\lambda^{k-1}, w^k, v^k))) \\
&\quad + \nabla\bar{F}(\lambda^k, w^{k+1}, v^{k+1}) - \bar{\nabla}\bar{F}(\lambda^k, w^{k+1}, v^{k+1})\|^2 \\
&\leq (1-\eta_k)^2 \mathbb{E}\|d^k - \bar{\nabla}\bar{F}(\lambda^{k-1}, w^k, v^k)\|^2 \\
&\quad + 2\eta^2 \mathbb{E}\|\nabla\bar{F}(\lambda^k, w^{k+1}, v^{k+1}) - \bar{\nabla}\bar{F}(\lambda^k, w^{k+1}, v^{k+1})\| \\
&\quad + 2(1-\eta_k)^2 \mathbb{E}\|\bar{\nabla}\bar{F}(\lambda^{k-1}, w^k, v^k) - \nabla\bar{F}(\lambda^{k-1}, w^k, v^k) \\
&\quad + \nabla\bar{F}(\lambda^k, w^{k+1}, v^{k+1}) - \bar{\nabla}\bar{F}(\lambda^k, w^{k+1}, v^{k+1})\|^2 \\
&\leq (1-\eta_k)^2 \mathbb{E}\|d^k - \bar{\nabla}\bar{F}(\lambda^{k-1}, w^k)\|^2 + \eta_k^2 \mathbb{E}\|\nabla\bar{F}(\lambda^k, w^{k+1}) - \bar{\nabla}\bar{F}(\lambda^k, w^{k+1}, v^{k+1}, v^{k+1})\|^2 \\
&\quad + (1-\eta_k)^2 \mathbb{E}_{\xi^k, \eta^k | \mathcal{M}^k/\{\xi^k, \eta^k\}} \|\bar{\nabla}\bar{F}(\lambda^{k-1}, w^k, v^k) - \bar{\nabla}\bar{F}(\lambda^k, w^{k+1})\|^2 \\
&\leq (1-\eta_k)^2 \|(d^k - \bar{\nabla}\bar{F}(\lambda^{k-1}, w^k, v^k)\|^2 + \eta_k^2 \left((\bar{p}_\lambda)^2 \sigma^2 + \underline{p}_\lambda \underline{p}_w \sigma_h^2\right) \\
&\quad + (1-\eta_k)^2 \mathbb{E}\|\bar{\nabla}\bar{F}(\lambda^{k-1}, w^k, v^k) - \bar{\nabla}\bar{F}(\lambda^k, w^{k+1}, v^{k+1})\|^2
\end{aligned}
\tag{41}
$$

where the second inequality uses Lemma 4 and Assumption 4, the last inequality uses the fact that $\mathbb{E}\|X\|^2 \geq \mathbb{E}\|X - \mathbb{E}X\|^2$. For the last term in the above inequality, using the definition $\bar{\nabla}\bar{F}(\lambda^{k-1}, w^k)$, it holds that

$$
\begin{aligned}
&\mathbb{E}\|\bar{\nabla}\bar{F}(\lambda^{k-1}, w^k, v^k) - \bar{\nabla}\bar{F}(\lambda^k, w^{k+1}, v^{k+1})\|^2 \\
&= 2\mathbb{E}\|\overline{m}_\lambda^k \nabla_\lambda F(\overline{m}_\lambda^{k-1}\lambda^{k-1}, \overline{m}_w w^k) - \overline{m}_\lambda^k \nabla_\lambda F(\overline{m}_\lambda^k \lambda^k, \overline{m}_w w^{k+1})\|^2 \\
&\quad + 2\mathbb{E}\|\underline{m}_{\lambda^k} \nabla_{\lambda w}^2 G(\underline{m}_\lambda^{k-1}\lambda^{k-1}, \underline{m}_w^k w^k)\underline{m}_w^k v^k - \underline{m}_{\lambda^k} \nabla_{\lambda w}^2 G(\underline{m}_\lambda^k \lambda^k, \underline{m}_w^k w^{k+1})\underline{m}_w^k v^{k+1}\|^2 \\
&\leq \left(2\overline{p}_\lambda (L_{12}^F)^2 + 2\underline{p}_\lambda \underline{p}_w r^2\right)\left(\mathbb{E}\|\lambda^{k-1} - \lambda^k\| + \mathbb{E}\|w^{k+1} - w^k\|^2\right) + 2\underline{p}_\lambda \underline{p}_w C_G^2 \mathbb{E}\|v^k - v^{k+1}\|^2,
\end{aligned}
\tag{42}
$$

Note that

$$
\begin{aligned}
&\mathbb{E}\|w^{k+1} - w^k\|^2 \\
&\leq 3\mathbb{E}\|w^k - w(\lambda^{k-1})\|^2 + 3\mathbb{E}\|w^{k+1} - w(\lambda^k)\|^2 + 3L_w^2 \mathbb{E}\|\lambda^k - \lambda^{k-1}\|^2 \\
&= 3\mathbb{E}\|w^k - w(\lambda^{k-1})\|^2 + 3\mathbb{E}\|w^{k+1} - w(\lambda^k)\|^2 + 3L_w^2 \alpha_{k-1}^2 \mathbb{E}\|d^{k-1}\|^2.
\end{aligned}
\tag{43}
$$

On the other hand, using the definition of $\tilde{v}^k$ and $v^k$, Lemma 6 and Proposition 2 we have that

$$
\mathbb{E}\|v^k - v^{k+1}\|^2 \leq 3V^k + 3V^{k+1} + 3C_w \alpha_k^2 \mathbb{E}\|d^{k-1}\|^2.
\tag{44}
$$

Combining (41), (42), (43) and (44), we have

$$
\begin{aligned}
&\mathbb{E}\|d^{k+1} - \bar{\nabla}\bar{F}(\lambda^k, w^{k+1}, v^{k+1})\|^2 \\
&\leq (1-\eta_k)^2 \|(d^k - \bar{\nabla}\bar{F}(\lambda^{k-1}, w^k, v^k)\|^2 + \eta_k^2 \left((\bar{p}_\lambda)^2\sigma^2 + \underline{p}_\lambda \underline{p}_w \sigma_h^2\right) \\
&+ (1-\eta_k)^2 \bar{p}_\lambda \left(2(L_{12}^F)^2 + 2r^2(1+3L_w^2) + 2C_G^2 3C_w\right) \alpha_{k-1}^2 \mathbb{E}\|d^{k-1}\|^2 \\
&+ (1-\eta_k)^2 \underline{p}_\lambda \left(2(L_{12}^F)^2 + 2r^2\right)\left(3\mathbb{E}\|w^k - w(\lambda^{k-1})\|^2 + 3\mathbb{E}\|w^{k+1} - w(\lambda^k)\|^2\right) \\
&+ 2(1-\eta_k)^2 \underline{p}_\lambda \underline{p}_w C_G^2 \left(3V^k + 3V^{k+1}\right) \\
&\leq (1-\eta_k)\|(d^k - \bar{\nabla}\bar{F}(\lambda^{k-1}, w^k, v^k)\|^2 + \eta_k^2 \left((\bar{p}_\lambda)^2\sigma^2 + \underline{p}_\lambda \underline{p}_w \sigma_h^2\right) \\
&+ \bar{p}_\lambda \left(2(L_{12}^F)^2 + 2r^2(1+3L_w^2) + 2C_G^2 3C_w\right) \alpha_{k-1}^2 \mathbb{E}\|d^{k-1}\|^2 \\
&+ \underline{p}_\lambda \left(2(L_{12}^F)^2 + 2r^2\right)\left(3\mathbb{E}\|w^k - w(\lambda^{k-1})\|^2 + 3\mathbb{E}\|w^{k+1} - w(\lambda^k)\|^2\right) \\
&+ 2\underline{p}_\lambda \underline{p}_w C_G^2 \left(3V^k + 3V^{k+1}\right).
\end{aligned}
\tag{45}
$$

Rearranging terms in the above inequality, we have

$$
\begin{aligned}
&\mathbb{E}\|d^{k+1} - \bar{\nabla}\bar{F}(\lambda^k, w^{k+1}, v^{k+1})\|^2 - \mathbb{E}\|(d^k - \bar{\nabla}\bar{F}(\lambda^{k-1}, w^k, v^k)\|^2 \\
&\leq -\eta_k \mathbb{E}\|(d^k - \bar{\nabla}\bar{F}(\lambda^{k-1}, w^k, v^k)\|^2 + \eta_k^2 \left((\bar{p}_\lambda)^2\sigma^2 + \underline{p}_\lambda \underline{p}_w \sigma_h^2\right) \\
&+ \bar{p}_\lambda \left(2(L_{12}^F)^2 + 2r^2(1+3L_w^2) + 2C_G^2 3C_w\right) \alpha_{k-1}^2 \mathbb{E}\|d^{k-1}\|^2 \\
&+ \underline{p}_\lambda \left(2(L_{12}^F)^2 + 2r^2\right)\left(3\mathbb{E}\|w^k - w(\lambda^{k-1})\|^2 + 3\mathbb{E}\|w^{k+1} - w(\lambda^k)\|^2\right) \\
&+ 2\underline{p}_\lambda \underline{p}_w C_G^2 \left(3V^k + 3V^{k+1}\right).
\end{aligned}
\tag{46}
$$

Now we bound the second term in (40). Using the definition of $\bar{\nabla}\bar{F}(\lambda^k, w^{k+1})$ and $\nabla F_{M^k}(\lambda^k)$, we have

$$
\begin{aligned}
&\mathbb{E}\|\bar{\nabla}\bar{F}(\lambda^k, w^{k+1}) - \nabla F_{M^k}(\lambda^k)\|^2 \\
&\leq \mathbb{E}\|\overline{m}_\lambda^k \nabla_\lambda F(\overline{m}_\lambda^k \lambda^k, \overline{m}_w w^{k+1}) - \overline{m}_\lambda^k \nabla_\lambda F(\overline{m}_\lambda^k \lambda^k, \overline{m}_w^k w(\lambda^k))\|^2 \\
&+ \mathbb{E}\|\underline{m}_{\lambda^k} \nabla_{\lambda w}^2 G(\underline{m}_\lambda^k \lambda^k, \underline{m}_w^k w^{k+1})\underline{m}_w^k v^{k+1} - \nabla_{\lambda w}^2 \tilde{G}(\lambda^k, w(\lambda^k))v_{\overline{m}_w^k, \lambda^k}\|^2 \\
&\leq \mathbb{E}\|\overline{m}_\lambda^k \nabla_\lambda F(\overline{m}_\lambda^k \lambda^k, \overline{m}_w w^{k+1}) - \overline{m}_\lambda^k \nabla_\lambda F(\overline{m}_\lambda^k \lambda^k, \overline{m}_w^k w(\lambda^k))\|^2 \\
&+ \mathbb{E}\|\underline{m}_{\lambda^k} \nabla_{\lambda w}^2 G(\underline{m}_\lambda^k \lambda^k, \underline{m}_w^k w^{k+1})\underline{m}_w^k \left(v^{k+1} - v_{\overline{m}_w^k, \lambda^k}\right)\|^2 \\
&+ \mathbb{E}\|\left(\underline{m}_{\lambda^k} \nabla_{\lambda w}^2 G(\underline{m}_\lambda^k \lambda^k, \underline{m}_w^k w^{k+1})\underline{m}_w^k - \nabla_{\lambda w}^2 \tilde{G}(\lambda^k, w(\lambda^k))\right) v_{\overline{m}_w^k, \lambda^k}\|^2 \\
&\leq \mathbb{E}\|\overline{m}_\lambda^k \nabla_\lambda F(\overline{m}_\lambda^k \lambda^k, \overline{m}_w w^{k+1}) - \overline{m}_\lambda^k \nabla_\lambda F(\overline{m}_\lambda^k \lambda^k, \overline{m}_w^k w(\lambda^k))\|^2 \\
&+ \mathbb{E}\|\underline{m}_{\lambda^k} \nabla_{\lambda w}^2 G(\underline{m}_\lambda^k \lambda^k, \underline{m}_w^k w^{k+1})\underline{m}_w^k \left(v^{k+1} - v_{\overline{m}_w^k, \lambda^k}\right)\|^2 \\
&+ \mathbb{E}\|\left(\underline{m}_{\lambda^k} \nabla_{\lambda w}^2 G(\underline{m}_\lambda^k \lambda^k, \underline{m}_w^k w^{k+1})\underline{m}_w^k - \underline{m}_{\lambda^k} \nabla_{\lambda w}^2 G(\underline{m}_\lambda^k \lambda^k, \underline{m}_w^k w(\lambda^k))\underline{m}_w^k\right) v_{\overline{m}_w^k, \lambda^k}\|^2 \\
&+ \mathbb{E}\|\left(\underline{m}_{\lambda^k} \nabla_{\lambda w}^2 G(\underline{m}_\lambda^k \lambda^k, \underline{m}_w^k w(\lambda^k))\underline{m}_w^k - \nabla_{\lambda w}^2 \tilde{G}(\lambda^k, w(\lambda^k))\right) v_{\overline{m}_w^k, \lambda^k}\|^2.
\end{aligned}
$$

Using Lemma 4, (23) together with Assumption 2, 3 and 4, the above inequality can be further passed to

$$
\begin{aligned}
&\mathbb{E}\|\bar{\nabla}\bar{F}(\lambda^k) - \nabla F_{\overline{m}_\lambda^k}(\lambda^k)\|^2 \\
&\leq \mathbb{E}\|\overline{m}_\lambda^k \nabla_\lambda F(\overline{m}_\lambda^k \lambda^k, \overline{m}_w w^{k+1}) - \overline{m}_\lambda^k \nabla_\lambda F(\overline{m}_\lambda^k \lambda^k, \overline{m}_w^k w(\lambda^k))\|^2 \\
&+ \bar{p}_\lambda \bar{p}_w n C_G^2 \mathbb{E}_{\underline{m}_w^k | \mathcal{M}^{k-1}} \mathbb{E}\|\underline{m}_w^k \left(v^{k+1} - E_{m^k} v_{\overline{m}_w^k, \lambda^k}\right)\|^2 \\
&+ \bar{p}_\lambda \bar{p}_w n L_{G^w} \mathbb{E}\|w^k - w(\lambda^k)\|\|E_{m^k} v_{\overline{m}_w^k, \lambda^k}\|^2.
\end{aligned}
$$

Using Lemma 6, Lemma 4 and the fact that $\underline{m}_w^k$ is a diagonal matrix with all entries belongs to $\{0, 1\}$, the above inequality can be further passed to

$$
\begin{aligned}
&\mathbb{E}\|\bar{\nabla}\bar{F}(\lambda^k) - \nabla F_{\overline{m}_\lambda^k}(\lambda^k)\|^2 \\
&\leq \left((L_{12}^F)^2\bar{p}_w + \bar{p}_\lambda\bar{p}_w n L_{G^w}^2 r^2\right)\mathbb{E}\|w^{k+1} - w(\lambda^k)\|^2 + \bar{p}_\lambda\bar{p}_w n C_G^2\mathbb{E}\|v^{k+1} - v_{\overline{m}_w^k,\lambda^k}\|^2 \quad (47) \\
&\leq \bar{p}_w\left((L_{12}^F)^2 + n L_{G^w}^2 r^2\right)\mathbb{E}\|w^{k+1} - w(\lambda^k)\|^2 + \bar{p}_\lambda\bar{p}_w n C_G^2 V^{k+1}
\end{aligned}
$$

Summing (38), $\frac{\alpha_k}{2}$ times of (40), $\alpha_k$ times of (47) and $\Gamma_3$ times of (46), we have that

$$
\begin{aligned}
&\mathbb{E}F_{M^{k+1}}(\lambda^{k+1}) - \mathbb{E}F_{M^k}(\lambda^k) + \Gamma_3\left(\mathbb{E}\|d^{k+1} - \bar{\nabla}\bar{F}(\lambda^k, w^{k+1}, v^{k+1})\|^2 - \mathbb{E}\|(d^k - \bar{\nabla}\bar{F}(\lambda^{k-1}, w^k, v^k)\|^2\right) \\
&\leq -\frac{\alpha_k}{2}\mathbb{E}\|\nabla F_{M^k}(\lambda^k)\|^2 \\
&\quad - \Gamma_3\alpha_k\eta_k\mathbb{E}\|(d^k - \bar{\nabla}\bar{F}(\lambda^{k-1}, w^k, v^k)\|^2 + \Gamma_3\alpha_k\eta_k^2\left((\bar{p}_\lambda)^2\sigma^2 + \underline{p}_\lambda\underline{p}_w\sigma_h^2\right) \\
&\quad + \Gamma_3\alpha_k\bar{p}_\lambda\left(2(L_{12}^F)^2 + 2r^2(1 + 3L_w^2) + 2C_G^2 3C_w\right)\alpha_{k-1}^2\mathbb{E}\|d^{k-1}\|^2 \\
&\quad + \Gamma_3\alpha_k\underline{p}_\lambda\left(2(L_{12}^F)^2 + 2r^2\right)\left(3\mathbb{E}\|w^k - w(\lambda^{k-1})\|^2 + 3\mathbb{E}\|w^{k+1} - w(\lambda^k)\|^2\right) \\
&\quad + 2\Gamma_3\alpha_k\underline{p}_\lambda\underline{p}_w C_G^2\left(3V^k + 3V^{k+1}\right) \\
&\quad + \alpha_k\bar{p}_w\left((L_{12}^F)^2 + n L_{G^w}^2 r^2\right)\mathbb{E}\|w^{k+1} - w(\lambda^k)\|^2 + \alpha_k\bar{p}_\lambda\bar{p}_w n C_G^2 V^{k+1} \\
&\quad + \left(\frac{\alpha_k^2 L_{\bar{F}}}{2} - \frac{\alpha_k}{2}\right)\mathbb{E}\|d^{k+1}\|^2.
\end{aligned}
$$

This completes the proof. $\qquad\square$

Now we present the detailed version of Theorem 2.

**Theorem 4.** *Consider (9). Suppose Assumptions 2, 3 and 4 hold. Let $\{(w^k, \lambda^k)\}$ be generated by Algorithm 1. Suppose $F$ is bounded below by $\bar{F}$. Then there exist small $\delta$ in Algorithm 1 such that*

$$
\begin{aligned}
&\frac{1}{T-1}\sum_{k=1}^{T-1}\mathbb{E}\|\nabla F_{M^k}(\lambda^k)\|^2 \leq \frac{1}{\sqrt{T}}\frac{1}{\underline{p}_w\underline{p}_\lambda}I_0 \\
&+ \frac{\ln(T+1)}{\sqrt{T}}O\left(\frac{\bar{p}_\lambda^2}{\underline{p}_\lambda\underline{p}_w} + \underline{p}_w\bar{p}_w\right)(\sigma^2 + \sigma_h^2) + \frac{\ln(T+1)}{\sqrt{T}}O\left(\underline{p}_w\left(\bar{p}_w^2(1 - \underline{p}_w^2) + \bar{p}_w(1 - \underline{p}_w)\right)\right)
\end{aligned}
$$
(48)

*Proof.* Rearrange (37) and summing it from 1 to $T$, we have

$$
\begin{aligned}
&\sum_1^T \frac{\alpha_k}{2}\mathbb{E}\|\nabla F_{M^k}(\lambda^k)\|^2 \leq -\mathbb{E}F_{M^{T+1}}(\lambda^{T+1}) + \mathbb{E}F_{M^0}(\lambda^0) \\
&+ \Gamma_3\left(\mathbb{E}\|(d^1 - \bar{\nabla}\bar{F}(\lambda^0, w^1, v^1)\|^2 - \mathbb{E}\|d^{T+1} - \bar{\nabla}\bar{F}(\lambda^k, w^{T+1}, v^{T+1})\|^2\right) \\
&- \Gamma_3\sum_1^T \alpha_k\eta_k\mathbb{E}\|(d^k - \bar{\nabla}\bar{F}(\lambda^{k-1}, w^k, v^k)\|^2 + \Gamma_3\alpha_k\eta_k^2\left((\bar{p}_\lambda)^2\sigma^2 + \underline{p}_\lambda\underline{p}_w\sigma_h^2\right) \\
&+ \sum_1^{T+1}\left(\Gamma_3\Gamma_d\alpha_k\alpha_{k-1}^2 + \frac{\alpha_k^2 L_{\bar{F}}}{2} - \frac{\alpha_k}{2}\right)\mathbb{E}\|d^{k-1}\|^2 \\
&+ \Gamma_w\sum_1^{T+1}\alpha_k\mathbb{E}\|w^k - w(\lambda^{k-1})\|^2 + 4\Gamma_3\sum_1^{T+1}\alpha_k\underline{p}_\lambda\underline{p}_w C_G^2 3V^k,
\end{aligned}
$$

where $\Gamma_v := 4\Gamma_3\underline{p}_\lambda\underline{p}_w C_G^2 3 + \bar{p}_\lambda\bar{p}_w n C_G^2$, $\Gamma_w := 2\Gamma_3\underline{p}_\lambda\left(2(L_{12}^F)^2 + 2r^2\right)3 + \bar{p}_w\left((L_{12}^F)^2 + n L_{G^w}^2 r^2\right)$, and $\Gamma_d := \Gamma_3\bar{p}_\lambda\left(2(L_{12}^F)^2 + 2r^2(1 + 3L_w^2) + 2C_G^2 3C_w\right)$. Summing

the above inequality with $\Gamma_2$ times of (30), we have

$$\sum_1^T \frac{\alpha_k}{2}\mathbb{E}\|\nabla F_{M^k}(\lambda^k)\|^2 + \Gamma_2\left(V^{T+2} - V^1\right)$$

$$\leq -\mathbb{E}F_{M^{T+1}}(\lambda^{T+1}) + \mathbb{E}F_{M^0}(\lambda^0)$$

$$+ \Gamma_3\left(\mathbb{E}\|(d^1 - \bar{\nabla}\bar{F}(\lambda^0, w^1, v^1)\|^2 - \mathbb{E}\|d^{T+1} - \bar{\nabla}\bar{F}(\lambda^k, w^{T+1}, v^{T+1})\|^2\right)$$

$$- \Gamma_3\sum_1^T \alpha_k\eta_k\mathbb{E}\|(d^k - \bar{\nabla}\bar{F}(\lambda^{k-1}, w^k, v^k)\|^2 + \Gamma_3\alpha_k\eta_k^2\left((\bar{p}_\lambda)^2\sigma^2 + \underline{p}_\lambda\underline{p}_w\sigma_h^2\right)$$

$$+ \sum_1^{T+1}\left(\Gamma_3\Gamma_d\alpha_k\alpha_{k-1}^2 + \frac{\alpha_k^2 L_{\bar{F}}}{2} - \frac{\alpha_k}{2} + \Gamma_2\frac{2}{\beta\underline{p}_w\mu}C_w\alpha_k\right)\mathbb{E}\|d^{k-1}\|^2$$

$$+ \sum_1^{T+1}\left(\Gamma_w\alpha_k + \Gamma_2\frac{2}{\underline{p}_w\mu}\beta_k L_{ww}^F\bar{p}_w\right)\mathbb{E}\|w^k - w(\lambda^{k-1})\|^2$$

$$+ \sum_1^{T+1}\left(\Gamma_2\left(-\frac{1}{2} + \frac{1}{4}\beta_k\underline{p}_w\mu\right)\underline{p}_w\mu\beta_k + 4\Gamma_3\underline{p}_\lambda\underline{p}_w C_G^2 3\alpha_k\right)V^k$$

$$+ \Gamma_2\sum_{k=1}^{T+1}\beta_k^2\left(4\bar{p}_w\sigma^2 + 4\bar{p}_w^2\sigma_h^2 + 4\bar{p}_w^2(1 - \underline{p}_w^2)nC_G^2 + 4\bar{p}_w(1 - \underline{p}_w)n(C^F)^2\right)$$

Summing this inequality with $\Gamma_1$ times of (26), we have

$$\sum_1^T \frac{\alpha_k}{2}\mathbb{E}\|\nabla F_{M^k}(\lambda^k)\|^2 + \Gamma_2\left(V^{T+2} - V^1\right) + \Gamma_1\left(\mathbb{E}\|w^{T+2} - w(\lambda^{T+1})\|^2 - \mathbb{E}\|w^1 - w(\lambda^0)\|^2\right)$$

$$\leq -\mathbb{E}F_{M^{T+1}}(\lambda^{T+1}) + \mathbb{E}F_{M^0}(\lambda^0)$$

$$+ \Gamma_3\left(\mathbb{E}\|(d^1 - \bar{\nabla}\bar{F}(\lambda^0, w^1, v^1)\|^2 - \mathbb{E}\|d^{T+1} - \bar{\nabla}\bar{F}(\lambda^k, w^{T+1}, v^{T+1})\|^2\right)$$

$$- \Gamma_3\sum_1^T \alpha_k\eta_k\mathbb{E}\|(d^k - \bar{\nabla}\bar{F}(\lambda^{k-1}, w^k, v^k)\|^2$$

$$+ \sum_1^{T+1}\left(\Gamma_3\Gamma_d\alpha_k\alpha_{k-1}^2 + \frac{\alpha_k^2 L_{\bar{F}}}{2} - \frac{\alpha_k}{2} + \Gamma_2\frac{2}{\beta\underline{p}_w\mu}C_w\alpha_k + \Gamma_1\frac{2(1 - \gamma_k\underline{p}_w\mu)}{\gamma\underline{p}_w\mu}L_w^2\alpha_k\right)\mathbb{E}\|d^{k-1}\|^2$$

$$+ \sum_1^{T+1}\left(\Gamma_w\alpha_k + \Gamma_2\frac{2}{\underline{p}_w\mu}\beta_k L_{ww}^F\bar{p}_w - \Gamma_1\frac{1}{2}\gamma_k\underline{p}_w\mu\right)\mathbb{E}\|w^k - w(\lambda^{k-1})\|^2$$

$$+ \sum_{k=1}^{T+1}\left(\Gamma_3\alpha_k\eta_k^2\bar{p}_\lambda^2 + \Gamma_1\gamma_k^2\bar{p}_w + \Gamma_2\beta_k^2 4\bar{p}_w\right)\sigma^2$$

$$+ \sum_1^{T+1}\left(\Gamma_2\left(-\frac{1}{2}\underline{p}_w\mu + \frac{1}{4}\beta_k\underline{p}_w^2\mu^2\right)\beta_k + 4\Gamma_3\underline{p}_\lambda\underline{p}_w C_G^2 3\alpha_k\right)V^k$$

$$+ \sum_{k=1}^{T+1}\left(\Gamma_2\beta_k^2 4\bar{p}_w^2 + \Gamma_3\alpha_k\eta_k^2\underline{p}_\lambda\underline{p}_w\right)\sigma_h^2$$

$$+ \sum_{k=1}^{T+1}\Gamma_2\beta_k^2\left(4\bar{p}_w^2(1 - \underline{p}_w^2)nC_G^2 + 4\bar{p}_w(1 - \underline{p}_w)n(C^F)^2\right)$$

$$\tag{49}$$

Let $\Gamma_2 := \frac{\beta\underline{p}_w\mu}{16C_w}, \Gamma_1 := \frac{\gamma\underline{p}_w\mu}{16L_w^2}$ be such that $-\frac{\alpha_k}{2} + \Gamma_2\frac{2}{\beta\underline{p}_w\mu}C_w\alpha_k + \Gamma_1\frac{2(1 - \gamma_k\underline{p}_w\mu)}{\gamma\underline{p}_w\mu}L_w^2\alpha_k <$ 0. Then there exists small $\delta$ with $\alpha_k = \frac{\delta}{\sqrt{T}}$ such that $\Gamma_3\Gamma_d\alpha_k\alpha_{k-1}^2 + \frac{\alpha_k^2 L_{\bar{F}}}{2} - \frac{\alpha_k}{2} +$

$\Gamma_2 \frac{2}{\beta \underline{p}_w \mu} C_w \alpha_k + \Gamma_1 \frac{2(1 - \gamma_k \underline{p}_w \mu)}{\gamma \underline{p}_w \mu} L_w^2 \alpha_k \leq 0$. Let $\Gamma_3 := \frac{1}{24 \underline{p}_\lambda \underline{p}_w C_G^2}$ and $\beta$ be small enough such that $\Gamma_2 \left( -\frac{1}{2} \underline{p}_w \mu + \frac{1}{4} \beta_k \underline{p}_w^2 \mu^2 \right) \beta_k + 4\Gamma_3 \underline{p}_\lambda \underline{p}_w C_G^2 3\alpha_k \leq 0$. Let $\gamma$ be big enough such that $\Gamma_w \alpha_k + \Gamma_2 \frac{2}{\underline{p}_w \mu} \beta_k L_{ww}^F \overline{p}_w - \Gamma_1 \frac{1}{2} \gamma_k \underline{p}_w \mu \leq 0$. Then (49) can be further passed to

$$
\sum_1^T \frac{\alpha_k}{2} \mathbb{E} \|\nabla F_{M^k}(\lambda^k)\|^2 + \Gamma_2 \left( V^{T+2} - V^1 \right) + \Gamma_1 \left( \mathbb{E}\|w^{T+2} - w(\lambda^{T+1})\|^2 - \mathbb{E}\|w^1 - w(\lambda^0)\|^2 \right)
$$

$$
\leq -\mathbb{E}F_{M^{T+1}}(\lambda^{T+1}) + \mathbb{E}F_{M^0}(\lambda^0) + \Gamma_3 \mathbb{E}\|(d^1 - \bar{\nabla}\bar{F}(\lambda^0, w^1, v^1)\|^2
$$

$$
+ \sum_{k=1}^{T+1} \left( \Gamma_3 \alpha_k \eta_k^2 \bar{p}_\lambda^2 + \Gamma_1 \gamma_k^2 \overline{p}_w + \Gamma_2 \beta_k^2 4\overline{p}_w \right) \sigma^2 + \sum_{k=1}^{T+1} \left( \Gamma_2 \beta_k^2 4\bar{p}_w^2 + \Gamma_3 \alpha_k \eta_k^2 \underline{p}_\lambda \underline{p}_w \right) \sigma_h^2
$$

$$
+ \sum_{k=1}^{T+1} \Gamma_2 \beta_k^2 \left( 4\bar{p}_w^2 (1 - \underline{p}_w^2) n C_G^2 + 4\bar{p}_w (1 - \underline{p}_w) n (C^F)^2 \right)
$$

$$
\leq -\bar{F} + \mathbb{E}F_{M^0}(\lambda^0) + \Gamma_3 \mathbb{E}\|(d^1 - \bar{\nabla}\bar{F}(\lambda^0, w^1, v^1)\|^2
$$

$$
+ \sum_{k=1}^{T+1} \left( \Gamma_3 \alpha_k \eta_k^2 \bar{p}_\lambda^2 + \Gamma_1 \gamma_k^2 \overline{p}_w + \Gamma_2 \beta_k^2 4\overline{p}_w \right) \sigma^2 + \sum_{k=1}^{T+1} \left( \Gamma_2 \beta_k^2 4\bar{p}_w^2 + \Gamma_3 \alpha_k \eta_k^2 \underline{p}_\lambda \underline{p}_w \right) \sigma_h^2
$$

$$
+ \sum_{k=1}^{T+1} \Gamma_2 \beta_k^2 \left( 4\bar{p}_w^2 (1 - \underline{p}_w^2) n C_G^2 + 4\bar{p}_w (1 - \underline{p}_w) n (C^F)^2 \right)
$$

$$
\tag{50}
$$

where the second inequality uses the assumption that $F$ is lower bounded by $\bar{F}$. Rearranging the above inequality we have

$$
\sum_1^T \frac{\alpha_T}{2} \mathbb{E} \|\nabla F_{M^k}(\lambda^k)\|^2 \leq \sum_1^T \frac{\alpha_k}{2} \mathbb{E} \|\nabla F_{M^k}(\lambda^k)\|^2
$$

$$
\leq \Gamma_2 V^1 + \Gamma_1 \mathbb{E}\|w^1 - w(\lambda^0)\|^2 - \bar{F} + \mathbb{E}F_{M^0}(\lambda^0) + \Gamma_3 \mathbb{E}\|(d^1 - \bar{\nabla}\bar{F}(\lambda^0, w^1, v^1)\|^2
$$

$$
+ \sum_{k=1}^{T+1} \left( \Gamma_3 \alpha_k \eta_k^2 \bar{p}_\lambda^2 + \Gamma_1 \gamma_k^2 \overline{p}_w + \Gamma_2 \beta_k^2 4\overline{p}_w \right) \sigma^2 + \sum_{k=1}^{T+1} \left( \Gamma_2 \beta_k^2 4\bar{p}_w^2 + \Gamma_3 \alpha_k \eta_k^2 \underline{p}_\lambda \underline{p}_w \right) \sigma_h^2
$$

$$
+ \sum_{k=1}^{T+1} \Gamma_2 \beta_k^2 \left( 4\bar{p}_w^2 (1 - \underline{p}_w^2) n C_G^2 + 4\bar{p}_w (1 - \underline{p}_w) n (C^F)^2 \right)
$$

$$
\tag{51}
$$

Using the definition of $w_1$ in Algorithm 1, we have that

$$
\mathbb{E}\|w^1\|^2 = \mathbb{E}\|w^0 - \gamma_k \underline{m}_w^0 \nabla_w G(\underline{m}_\lambda^0 \lambda^0, \underline{m}_w^0 w^0; \xi^0)\|
$$

$$
\leq 4\|w^0\|^2 + 4\mathbb{E}\|\gamma_k \underline{m}_w^0 \nabla_w G(\lambda^0, w^0)\|^2
$$

$$
+ 4\mathbb{E}\|\gamma_k \underline{m}_w^0 \left( \nabla_w G(\underline{m}_\lambda^0 \lambda^0, \underline{m}_w^0 w^0) - \nabla_w G(\lambda^0, w^0) \right)\|^2 + \gamma_k^2 \sigma^2
$$

$$
\leq 4\|w^0\|^2 + 4\mathbb{E}\|\gamma_k \underline{m}_w^0 \nabla_w G(\lambda^0, w^0)\|^2
$$

$$
+ 4\mathbb{E}\|\gamma_k \left( \nabla_w G(\underline{m}_\lambda^0 \lambda^0, \underline{m}_w^0 w^0) - \nabla_w G(\lambda^0, w^0) \right)\|^2 + \gamma_k^2 \sigma^2
$$

$$
\leq 4\|w^0\|^2 + 4\gamma_k^2 \overline{p}_w \|\nabla_w G(\lambda^0, w^0)\|^2 + 4\mathbb{E}\gamma_k^2 L_G^2 \|(\underline{m}_\lambda^0 \lambda^0, \underline{m}_w^0 w^0) - (\lambda^0, w^0)\|^2 + \gamma_k^2 \sigma^2
$$

$$
\leq 4\|w^0\|^2 + 4\gamma_k^2 \overline{p}_w \|\nabla_w G(\lambda^0, w^0)\|^2 + 4\gamma_k^2 L_G^2 \|(\lambda^0, w^0)\|^2 + \gamma_k^2 \sigma^2.
$$

Thus,

$$
\mathbb{E}\|w^1 - w(\lambda^0)\|^2 \leq 8\|w^0\|^2 + 8\gamma_0^2 \overline{p}_w \|\nabla_w G(\lambda^0, w^0)\|^2
$$

$$
+ 8\gamma_0^2 L_G^2 \|(\lambda^0, w^0)\|^2 + 2\gamma_0^2 \sigma^2 + 2\|w(\lambda^0)\|^2
$$

$$
\tag{52}
$$

For the term $V^1$, we first bound $\|v^1\|$. By definition,

$$
\begin{aligned}
&\|v^1\| \\
&\leq 3\mathbb{E}\|\beta_k\overline{m}_w^0\nabla_w F(\overline{m}_\lambda^0\lambda^0, \overline{m}_w^0 w^1; \eta^0)\|^2 + 3\mathbb{E}\|\underline{m}_w^0 v^0\|^2 + \mathbb{E}\|\beta_k\underline{m}_w^0\nabla_{ww}^2 G(\lambda^0, w^0; \xi^0)\underline{m}_w^0 v^0\|^2 \\
&\leq 3\mathbb{E}\|\nabla_w F(\overline{m}_\lambda^0\lambda^0, \overline{m}_w^0 w^1; \eta^0)\|^2 + 3\|v^0\|^2 + 3\|\beta_k\nabla_{ww}^2 G(\lambda^0, w^0; \xi^0)v^0\|^2 \\
&\leq 6\mathbb{E}\|\nabla_w F(\overline{m}_\lambda^0\lambda^0, \overline{m}_w^0 w^1)\|^2 + 6\sigma^2 + 3\|v^0\|^2 + 6\|\beta_k\nabla_{ww}^2 G(\lambda^0, w^0)v^0\|^2 + 6n\sigma_h^2 \\
&\quad + 2\frac{1}{\underline{p}_w\mu}\mathbb{E}\|\nabla_w F(\overline{m}_\lambda^0\lambda^0, \overline{m}_w^0 w(\lambda^0))\|^2 \\
&\leq 6(L_{12}^F)^2\|\lambda^0\|^2 + 6(L_{22}^F)^2\|w^1\|^2 + 6\sigma^2 + 3\|v^0\|^2 + 6\|\beta_k\nabla_{ww}^2 G(\lambda^0, w^0)v^0\|^2 + 12n\sigma_h^2
\end{aligned}
$$

On the other hand,

$$
\begin{aligned}
\mathbb{E}\|v_{\overline{m}_w^0, \lambda^0}\|^2 &\leq \mathbb{E}\|\left(\nabla_{ww}^2\tilde{G}(\lambda^0, w(\lambda^0))\right)^{-1}\overline{m}_w^0\nabla_w F(\overline{m}_\lambda^0\lambda^0, \overline{m}_w^0 w(\lambda^0))\|^2 \\
&\leq \frac{1}{\underline{p}_w\mu}\mathbb{E}\|\nabla_w F(\overline{m}_\lambda^0\lambda^0, \overline{m}_w^0 w(\lambda^0))\|^2 \leq \frac{1}{\underline{p}_w\mu}\left(12(L_{12}^F)^2\|\lambda^0\|^2 + 12(L_{22}^F)^2\|w(\lambda^0)\|^2\right).
\end{aligned}
$$

Thus,

$$
\begin{aligned}
V^1 &\leq 12(L_{12}^F)^2\|\lambda^0\|^2 + 12(L_{22}^F)^2\|w^1\|^2 + 12\sigma^2 + 6\|v^0\|^2 + 12\|\beta_k\nabla_{ww}^2 G(\lambda^0, w^0)v^0\|^2 \\
&\quad + 12n\sigma_h^2 + 2\frac{1}{\underline{p}_w\mu}\left(12(L_{12}^F)^2\|\lambda^0\|^2 + 12(L_{22}^F)^2\|w(\lambda^0)\|^2\right).
\end{aligned} \tag{53}
$$

Using (52), the above inequality can be further passed to

$$
\begin{aligned}
\mathbb{E}\|V_1\|^2 &\leq 12(L_{12}^F)^2\|\lambda^0\|^2 + 12\sigma^2 + 6\|v^0\|^2 + 12\|\beta_k\nabla_{ww}^2 G(\lambda^0, w^0)v^0\|^2 + 12n\sigma_h^2 \\
&\quad + 2\frac{1}{\underline{p}_w\mu}\left(12(L_{12}^F)^2\|\lambda^0\|^2 + 12(L_{22}^F)^2\|w(\lambda^0)\|^2\right) \\
&\quad + 12(L_{22}^F)^2\left(4\|w^0 - w(\lambda^0)\|^2 + 4\gamma_k^2\overline{p}_w\|\nabla_w G(\lambda^0, w^0)\|^2 + 4\gamma_k^2 L_G^2\|(\lambda^0, w^0)\|^2 + \gamma_k^2\sigma^2\right).
\end{aligned}
$$

As for $\mathbb{E}F_{M^0}(\lambda^0)$, using the definition of $F_M$ and Assumption 2, it holds that

$$
\|F_{M^0}(\lambda^0)\| \leq \|f(\lambda^0)\| + C^F\|\lambda^0\| = \|F(\lambda^0, w(\lambda^0))| \tag{54}
$$

Using the definition in (39)

$$
\begin{aligned}
\|\overline{\nabla}\overline{F}(\lambda^0, w^1, v^1)\|^2 &= \|\overline{m}_\lambda^1\nabla_\lambda F(\overline{m}_\lambda^0\lambda^0, \overline{m}_w w^1) - \underline{m}_{\lambda^0}\nabla_{\lambda w}^2 G(\underline{m}_\lambda^0\lambda^0, \underline{m}_w^0 w^1)\underline{m}_w^0 v^1\|^2 \\
&\leq L_F^2 + C_G^2 r^2
\end{aligned} \tag{55}
$$

and

$$
\begin{aligned}
\|d^1\|^2 &= \|\nabla\overline{F}(\lambda^0, w^1, v^1) + (1 - \eta_0)(d^0 - \nabla\overline{F}(\lambda^0, w^0, v^{-1})))\|^2 \\
&\leq \|\overline{m}_\lambda^k\nabla_\lambda F(\overline{m}_\lambda^0\lambda^0, \overline{m}_w w^1; \eta^0) - \underline{m}_{\lambda^0}\nabla_{\lambda w}^2 G(\underline{m}_\lambda^0\lambda^0, \underline{m}_w^0 w^1; \xi^0)\underline{m}_w^k v^1\| \\
&\quad + \|(1 - \eta_0)(d^0 - \nabla\overline{F}(\lambda^0, w^0, v^{-1})))\|^2 \\
&\leq (C^F)^2 + 2C_G^2 r^2 + \|(1 - \eta_0)(d^0 - \nabla\overline{F}(\lambda^0, w^0, v^{-1})))\|^2
\end{aligned} \tag{56}
$$

Combining (53), (54), (55), (56) , we know that there exists $I_0$ such that (58) ca be further passed to

$$\frac{1}{\sqrt{T}} \sum_1^T \frac{1}{2} \mathbb{E} \|\nabla F_{M^k}(\lambda^k)\|^2 \le \frac{1}{\underline{p}_w \underline{p}_\lambda} I_0$$

$$+ \sum_{k=1}^{T+1} \left(\Gamma_3 \alpha_k \eta_k^2 \bar{p}_\lambda^2 + \Gamma_1 \gamma_k^2 \overline{p}_w + \Gamma_2 \beta_k^2 4\bar{p}_w\right) \sigma^2 + \sum_{k=1}^{T+1} \left(\Gamma_2 \beta_k^2 4\bar{p}_w^2 + \Gamma_3 \alpha_k \eta_k^2 \underline{p}_\lambda \underline{p}_w\right) \sigma_h^2$$

$$+ \sum_{k=1}^{T+1} \Gamma_2 \beta_k^2 \left(4\bar{p}_w^2 (1 - \underline{p}_w^2) n C_G^2 + 4\bar{p}_w (1 - \underline{p}_w) n (C^F)^2\right) \tag{57}$$

$$\le \frac{1}{\underline{p}_w \underline{p}_\lambda} I_0 + \ln(T+1) \left(\Gamma_3 \delta \eta \bar{p}_\lambda^2 + \Gamma_1 \gamma^2 \delta^2 \overline{p}_w + \Gamma_2 \beta^2 \delta^2 4\bar{p}_w\right) \sigma^2$$

$$+ \ln(T+1) \left(\Gamma_2 \beta^2 \delta^2 4\bar{p}_w^2 + \Gamma_3 \delta \eta^2 \underline{p}_\lambda \underline{p}_w\right) \sigma_h^2$$

$$+ \ln(T+1) \Gamma_2 \beta^2 \delta^2 \left(4\bar{p}_w^2 (1 - \underline{p}_w^2) n C_G^2 + 4\bar{p}_w (1 - \underline{p}_w) n (C^F)^2\right).$$

Deviding both sides by $\sqrt{T}$, we have

$$\frac{1}{T} \sum_1^T \frac{1}{2} \mathbb{E} \|\nabla F_{M^k}(\lambda^k)\|^2$$

$$\le \frac{1}{\sqrt{T}} \frac{1}{\underline{p}_w \underline{p}_\lambda} I_0 + \frac{\ln(T+1)}{\sqrt{T}} \left(\Gamma_3 \delta \eta \bar{p}_\lambda^2 + \Gamma_1 \gamma^2 \delta^2 \overline{p}_w + \Gamma_2 \beta^2 \delta^2 4\bar{p}_w\right) \sigma^2$$

$$+ \ln(T+1) \left(\Gamma_2 \beta^2 \delta^2 4\bar{p}_w^2 + \Gamma_3 \delta \eta^2 \underline{p}_\lambda \underline{p}_w\right) \sigma_h^2$$

$$+ \frac{\ln(T+1)}{\sqrt{T}} \Gamma_2 \beta^2 \delta^2 \left(4\bar{p}_w^2 (1 - \underline{p}_w^2) n C_G^2 + 4\bar{p}_w (1 - \underline{p}_w) n (C^F)^2\right) \tag{58}$$

$$= \frac{1}{\sqrt{T}} \frac{1}{\underline{p}_w \underline{p}_\lambda} I_0$$

$$+ \frac{\ln(T+1)}{\sqrt{T}} O\left(\frac{\bar{p}_\lambda^2}{\underline{p}_\lambda \underline{p}_w} + \underline{p}_w \bar{p}_w\right) (\sigma^2 + \sigma_h^2)$$

$$+ \frac{\ln(T+1)}{\sqrt{T}} O\left(\underline{p}_w \left(\bar{p}_w^2 (1 - \underline{p}_w^2) + \bar{p}_w (1 - \underline{p}_w)\right)\right).$$

This completes the proofs. $\square$