# OpenReview forum: "Dropout Enhanced Bilevel Training"
_ICLR.cc/2024/Conference — ICLR 2024 spotlight_

### Official Review · Reviewer_mkUf · 2023-10-30

**Soundness:** 3 good
**Presentation:** 2 fair
**Contribution:** 2 fair
**Rating:** 6
**Confidence:** 4

**Summary:**

This paper points out that overfitting is a common challenge in bilevel training task. To address this issue, the author proposes the dropout mask for the bilevel optimization problem. Specifically, the author proves the convergence of dropout bilevel methods theoretically and empirically shows that proposed method mitigates the overfitting issue.

**Strengths:**

Overfitting issue has not been fully investigated in the bilevel area and this paper proposes the new method to address it.

**Weaknesses:**

1. It looks like this paper studies data cleaning problem rather than the bilevel problem. Both the introduction and experiments are based on data cleaning context. The authors are suggested to discuss and conduct experiments on other bilevel optimization settings to demonstrate applicability of proposed algorithm.

2. Overfitting issue can be easily addressed by early stopping. The authors are encouraged to make an experimental comparison between proposed dropout and early stopping to demonstrate the necessity of adopting dropout.

3. In Theorem 2, the authors demonstrate that dropout rate "influences" the upper bound. However, it only influences the constant term and does not influence the convergence rate. Furthermore, the theoretical results only demonstrates the dropout rates "influence" the convergence but does not demonstrates it "improves" the convergence rate, even for the constant term. I expect to see inspiration about dropout rate selection from this theorem. The same issue also occurs for the Lemma 1 which is about variance term.

**Questions:**

Can you discuss the technical difficulties of applying dropout analysis into bilevel optimization problem? That will be helpful to understand the theoretical contribution.

---

> ### Author Response · Authors · 2023-11-20
>
> **Responses to weakness section point 1:** Please see the global responses Section 2.
>
> **Responses to weakness section point 2:** Please refer to global responses Section 3.
>
> **Responses to weakness section point 3:** Thank you for pointing this out. In our analysis, the dropout rate indeed influences the convergence rate. In our initial analysis, we simplified the dropout rate in the convergence rate estimation to $1$. We have now refined the proof for this aspect in the revised paper to make the convergence analysis more precise. The constant in the convergence rate is expressed as $O(\frac{1}{\bar p_\lambda \bar p_w})$, where $\bar p_\lambda$ and $\bar p_w$ represent the maximum dropout rates related to the upper and lower level parameters, respectively. As can be seen, the convergence rate is lower when dropout is applied compared to when it is not. This observation is confirmed by our experimental results, particularly in Figure 2 in the revised version of the paper (Figure 4 in the original version).

---

> > ### Comment · Reviewer_mkUf · 2023-11-20
> >
> > Thank you for your response. It greatly addressed my concern and I raise my score to 6.

---

> > > ### Author Response · Authors · 2023-11-21
> > >
> > > We sincerely thank the reviewer for your valuable comments and appreciate the increase in scores.

---

### Official Review · Reviewer_fs9w · 2023-11-01

**Soundness:** 4 excellent
**Presentation:** 4 excellent
**Contribution:** 3 good
**Rating:** 8
**Confidence:** 3

**Summary:**

The paper uses dropout methods for bilevel training tasks. Bilevel optimization problems consist of two intertwined optimization problems and can be particularly sensitive to small changes, especially when data is limited. The study introduces a bilevel optimization model that considers the distribution of dropout masks and examines how varying dropout rates impact the hypergradient of this model. The authors adapt an existing bilevel method to incorporate dropout and provide theoretical convergence guarantees for this new approach. Empirical tests on data cleaning problems show that this method can mitigate overfitting.

**Strengths:**

Incorporating dropout in bilevel training tasks is a novel approach, offering a new method to combat overfitting in such tasks. The authors study the convergence properties of the introduced method. The study offers empirical proof, especially in the context of data cleaning problems, demonstrating the efficacy of the proposed method in reducing overfitting. The paper paves the way for adapting other state-of-the-art bilevel methods to account for dropout, making it a foundational study for further research in this direction.

**Weaknesses:**

The use of datasets like MNIST, which is relatively small and simplistic, might not fully showcase the potential or limitations of the proposed method in real-world, complex scenarios.

**Questions:**

Beyond data cleaning, how might this method be applied to other machine learning tasks, especially those that inherently involve bilevel optimization?

---

> ### Author Response · Authors · 2023-11-20
>
> Thanks for the suggestions. We now added more experiments on different bilevel problems. Please see the global responses Section 1.

---

### Official Review · Reviewer_hA9j · 2023-11-03

**Soundness:** 3 good
**Presentation:** 3 good
**Contribution:** 3 good
**Rating:** 8
**Confidence:** 3

**Summary:**

This paper brings dropout to bilevel optimization to avoid overfitting issues. The author forms a statistical bilevel optimization problem that includes the distribution of the dropout masks and propose a dropout method to solve it. To analyze the convergence of the proposed method, they delicately quantify the bias induced by the dropout. The empirical performance of the proposed method is tested on the data hyper cleaning task and shows that dropout efficiently improve the test accuracy and is more stable.

**Strengths:**

1. This paper incorporates the dropout technique in the bilevel optimization to avoid overfitting. The idea is new to the bilevel optimization community and has been tested effective.
2. The analysis of dropout by modeling the dropout mask as a stochastic distribution is thorough and novel.

Overall, I felt this is a solid and novel paper for bilevel optimization.

**Weaknesses:**

1. The motivation of considering the dropout in bilevel optimization could be explained more. For example, why avoiding overfitting is important in bilevel optimization, that is, does bilevel structure exacerbate the overfitting issues?
2. The convergence analysis is built up on the stationarity measure $\frac{1}{T-1} \sum_{k=1}^{T-1} \mathbb{E}\left\\|\nabla F_{M^{k}}\left(\lambda^{k}\right)\right\\|^{2}$, which is the dropout bilevel objective. To fortify the analysis, it is conceivable to draw connections between this measure and the one based on the original bilevel objective $\frac{1}{T-1} \sum_{k=1}^{T-1} \mathbb{E}\left\\|\nabla F\left(\lambda^{k}, w(\lambda^{k})\right)\right\\|^{2}$. This could elucidate the relaxation error introduced by dropout and establish a guarantee for the original objective's convergence, thereby validating efficiency of the dropout approach more rigorously.

**Questions:**

See weakness.

---

> ### Author Response · Authors · 2023-11-20
>
> **Responses to weakness point 1:** Thank you for your suggestions. We have explained more about the motivation of considering the dropout in bilevel optimization in the introduction section of our revised version. In the context of a data cleaning task, the goal is to train a classifier using corrupted training data. As evidenced by our experiments, overfitting to the training data leads to a decrease in the testing accuracy of the classifier. Therefore, addressing the overfitting problem is essential to enhance the classifier's generalization ability in data cleaning tasks.
>
> Regarding meta-learning, the goal is to develop models capable of rapidly learning new tasks or adapting to new environments with minimal data. This objective can be formulated as a bilevel optimization problem. Overfitting in the model learned through bilevel optimization can result in diminished ability to learn new tasks. For instance, in classification problems, a model that overfits the training data may fail to effectively classify new data after only a few training steps.
>
> **Responses to weakness point 2:** Please see the global responses Section 2.

---

### Official Review · Reviewer_Kt41 · 2023-11-11

**Soundness:** 3 good
**Presentation:** 3 good
**Contribution:** 3 good
**Rating:** 6
**Confidence:** 4

**Summary:**

This paper proposes a dropout method to address the issue of overfitting in bilevel training tasks. The authors provide theoretical convergence guarantees from an optimization perspective and demonstrate its effectiveness through experiments, using data cleaning as an illustrative example.

**Strengths:**

The paper is generally well-written and easy to follow. It investigates a relatively unexplored area and aims to analyze the dropout bilevel method for addressing this problem. Another advantage is the detailed theoretical analysis provided in the paper.

**Weaknesses:**

- Based on the results figures, it appears that early stopping can effectively resolve the issue of overfitting, even though the accuracy without dropout is higher.
- The theoretical analysis in the current context is not particularly challenging. In contrast, in many stochastic bilevel optimization studies, such as [2], a more comprehensive framework is presented, along with convergence rate guarantees for stochastic bilevel optimization. You can try to answer the extra challenge in these theoretical literatures [2].
- The experimentation conducted on data cleaning is insufficient, and it would be better to observe more results on other bilevel optimization tasks.
- Each figures should be accompanied by a brief caption.

**Questions:**

- In addition to the theoretical analysis, literature [1] also investigates the impact of dropout rate on bilevel optimization. However, what are the other distinguishing factors between these studies?
- What is the relationship between the objective function after incorporating dropout and the original objective function?

[1] Delta-STN: Efficient Bilevel Optimization for Neural Networks using Structured Response Jacobians. Juhan Bae, Roger Grosse. NeurIPS 2020.

[2] Bilevel Optimization: Convergence Analysis and Enhanced Design. Kaiyi Ji, Junjie Yang, Yingbin Liang. ICML 2021.

---

> ### Author Response · Authors · 2023-11-20
>
> **Responses to weakness point 1:** Thanks for the comments. Please refer to global responses Section 3.
>
> **Responses to weakness point 2:** Thanks for the comments. Please note that a significant analytical challenge in our work is addressing the distribution of dropout masks. Given the nested structure of bilevel optimization, the dropout masks at both the lower and upper levels contribute to the complexity of estimating convergence rates. The referenced study [2] analyzes the convergence rate of bilevel optimization methods using stochastic gradients. Our work also addresses the convergence analysis of methods employing stochastic gradients. In contrast to [2], our study not only addresses the convergence rate of stochastic methods with stochastic gradients but also tackles the randomness introduced by dropout.
>
> **Responses to weakness point 3:** We now added additional experiments for the application of meta-learning. Please see the results in the global responses Section 1.
>
> **Responses to weakness point 4:** Thanks for your suggestion. We have made all captions shorter.
>
> **Responses to Questions point 1:** There are three distinct differences between our study and [1]:
>
> **1. Problem Settings:** The primary difference between our work and [1] lies in the problem settings. In [1], the dropout rate is treated as a variable to optimize at the upper level. Conversely, in our settings, the dropout rate is a fixed hyperparameter, not a variable requiring optimization. The objective of [1] is to find the optimal dropout rate for the lower-level optimization problem, categorizing it as a hyperparameter optimization problem. Our bilevel optimization model encompasses not only hyperparameter optimization but also extends to other applications such as data cleaning problems and meta-learning.
>
> **2. Approaches to Bilevel Optimization:** The methodologies employed in [1] represent a different genre. [1] focuses on approximating the solution of the lower problem with a neural network, effectively altering the lower-level problem in the bilevel optimization framework. In contrast, our work belongs to the genre that endeavors to solve the original lower-level problem.
>
> **3.Theoretical Considerations:** Owing to the differing approaches to bilevel optimization, the theoretical concerns of our work and [1] vary significantly. [1] analyzes the accuracy of the network's approximation of the lower-level solution, whereas we discuss the convergence rate of the bilevel method that applies dropout.
>
> In summary, [1] fundamentally differs from our work in terms of the bilevel problems considered, the approaches used to solve these problems, and the resulting theoretical implications. We now include [1] as related work in the revised version.
>
>
> **Responses to Questions point 2:** Please see Section 2 of the global responses.

---

> ### Comment · Reviewer_Kt41 · 2023-11-21
>
> Thanks for your response. I still have several questions.
> - (Novelty) Using the dropout technique in the bilevel optimization to avoid overfitting seems not novel. The effectiveness of dropout in neural networks, especially with bi-level applications, is well-known and expected, but this confirmation does not deepen our understanding of the subject. Therefore, I think this paper does not merit a score of 8. The contribution is limited.
> - (Responses to weakness point 2) What is the difference in meaning between the terms "stochastic" and "randomness"?
> - (Responses to weakness point 4) I apologize for any confusion in my previous statement. I meant to convey that each figure should be accompanied by a brief explanatory text, rather than just a short title.
> - (Why not early stopping) In Figure 3, new experiments on meta-learning demonstrate the absence of an overfitting issue.

---

> > ### Author Response · Authors · 2023-11-22
> >
> > Thanks for your further comments. Here are our responses.
> >
> > **Responses to point 1:** The dropout method has been popularly used in deep neural networks, but this is totally DIFFERENT to our work. We focus on bilevel optimization problems and propose to use dropout in general bilevel optimization, which is not limited to neural network. Especially our paper makes novel theoretical contributions. We analyze the convergence rates of the dropout bilevel method. To the best of our knowledge, the current literature on the theoretical analysis of bilevel methods is limited to classical bilevel optimization models that have not considered the existence of dropout. Therefore, we are the first to break this limit.
> >
> > **Responses to point 2:** By 'stochastic methods,' we refer to the approach in [2] and other existing bilevel methods that use stochastic gradients. They are the gradients of $F$ and $G$ at randomly sampled training and validation data points.  In addition to the randomness introduced by sampling data points, the dropout bilevel method has the randomness of sampling dropout masks. The term 'stochastic' is an adjective and 'randomness' is a noun, yet both refer to the same mathematical concept.
> >
> > **Responses to point 3:** Thanks for pointing this out. We now added explanatory text in the caption of each figures in the revised version.
> >
> > **Responses to point 4:** We must clarify that the overfitting did occur. In meta-learning, overfitting does not show up as a decrease in testing accuracy. Instead, it appears as staying at a low testing accuracy. This can be seen in the first line of Figure 3, where the training accuracy converges to a high value towards the end of the training process, while the testing accuracy remains low. After the addition of dropout, there is a noticeable improvement in the testing accuracy. Nevertheless, the result of the data cleaning experiments in Figure 2 (c) is sufficient to show that the early stopping may not be a better choice to address the overfitting.

---

> > > ### Comment · Reviewer_Kt41 · 2023-11-22
> > >
> > > Thanks. You have addressed my concerns. I will raise my score to 6.

---

> > > > ### Author Response · Authors · 2023-11-22
> > > >
> > > > We sincerely appreciate the reviewer's valuable comments and are grateful for the increased score.

---

### Author Response · Authors · 2023-11-20

We thank the reviewers for their efforts in reviewing this paper. We address the common points mentioned by several reviewers in this global response.

**1. Additional experiments**


Suggestions from reviewers indicate that we should add more experiments, beyond data cleaning, to demonstrate the generalizability of the dropout bilevel method.

In response to this comment, we have applied our method to another type of bilevel problem: meta-learning, as discussed by Franceschi et al. [2018]. Specifically, we focus on few-shot learning, following the experimental protocols of Vinyals et al. [2016].
 The bilevel formulation is intrduced in (2) of the revised version of the work and the experiment details are added in Section 5. Figure 3 in the revised version shows the accuracy of the method when adding dropout at different rates to the $i$th layer. Figure 3 also shows the accuracy against the iteration for each training process, and shows adding appropriate dropout to any layer improves the testing accuracy.

**2. Relation between the dropout bilevel model and the original bilevel model**

It is natural to consider the relationship between the stationarity of the dropout bilevel optimization problem and that of the original bilevel optimization problem. We have found that the stationary point of the model employing dropout is not the same as that of the original one.

  Let's consider a simple 2-dimensional case. Let $f(x_1,x_2): = \frac12 x_1^2x_2 + 1000x_1$. Then $\nabla f(x) = (x_1x_2 + 1000,\frac12 x_1^2)$. Let  $m= diag(m_1,m_2)$ with $m_1 \approx Bernoulli(p)$ and $m_2 \approx Bernoulli(p)$.
  Then $\\mathbb{E}_m f(mx) = p^2f(x_1,x_2) + (1 - p)pf(0,x_2) + (1 - p)p f(x_1,0) + (1-p)^2f(0,0)$ and
$$
\nabla \mathbb{E}_m f(mx) = p^2(x_1x_2 + 1000,\frac12 x_1^2) + (1 - p)p(1000,0) + (1 - p)p ( 1000, \frac12 x_1^2)= p^2(x_1x_2 + 1000,\frac12 x_1^2) + (1 - p)p(2000, \frac12 x_1^2).
$$

Let $x^k=(x_1^k,x_2^k)$ be with $x_1^k = -\frac{1000p^2 + 2000(1-p)p}{k}$ and $x_2^k = k$. Then
${lim}_k\nabla \mathbb{E}_m f(mx^k)  =  0$. However, ${lim}_k\nabla f(x^k) = ( -1000p^2 - 2000(1-p)p+1000,0) $.
Therefore, the convergence of $\nabla \mathbb{E}_m f(mx^k)$ does not necessarily imply the convergence of $\nabla f(x^k)$. And the error between $\nabla f(x^k)$ and $\nabla \mathbb{E}_m f(mx^k)$ can not be closed unless $p$ approaches $1$. Therefore,  there are additional errors brought only by the dropout. This example implies that the dropout bilevel method may not optimize the original bilevel problem. This is expected because when the training loss is optimized, the model fits the training data too well and this will increase the chance of overfitting.

**3. Why not early stopping**

We would like to clarify early stopping cannot always resolve the issue of overfitting. As shown in Figure 2 (c) of the revised version of the paper (Figure 4 in the original version), the testing accuracy of the method with dropout (indicated by the red line) is significantly higher than the peak testing accuracy of the method without dropout (the green line). Additionally, our new experiments on another bilevel problem, as illustrated in Figure 3 of the revised version, demonstrate that the testing accuracy does not decrease when dropout is not applied. However, the application of proper dropout leads to an increase in testing accuracy. These findings suggest that early stopping may not always be a preferable alternative to dropout. Therefore, it is still worthwhile to explore the application of dropout.

---

### Meta-Review · Area_Chair_8YbZ · 2023-12-09

**Metareview:**

In order to mitigate overfitting in bilevel optimization, this paper studies dropout, both theoretically and empirically.
The main motivating application is data cleaning, framed as weighing of the training data points. The reviewers criticized this strong focus as not general enough, and during the rebuttal, the authors added an application in meta-learning. Overfitting materializes a lot less in this second application, but it can still be seen.
Strengths of the paper include clarity, the detailed theoretical analysis, and studying a relatively new aspect of bilevel optimization.
Weaknesses include too much focus on the data cleaning application (partially mitigated during the rebuttal by a second application, but this necessarily did not receive as much love as the first application) and not clear enough delineation from methods that already use dropout in bilevel optimization.
Overall, this is a good contribution to the field of bilevel optimization and I recommend acceptance.

**Justification For Why Not Higher Score:**

Too much focus on the data cleaning application (partially mitigated during the rebuttal by a second application, but this necessarily did not receive as much love as the first application)
Also, Dropout is very well established in deep learning and has also been studied in meta-learning; the paper cites some works but does not describe differences and does not spell out sufficiently in which way its treatment is different.

**Justification For Why Not Lower Score:**

The paper studies a relatively new aspect of bilevel optimization, includes a detailed theoretical analysis and shows strong empirical results.

---

### Decision · Program_Chairs · 2024-01-16

Accept (spotlight)